# Kernel Neural Operators (KNOs) for Scalable, Memory-efficient, Geometrically-flexible Operator Learning

**Matthew Lowery**                                                      *mlowery@cs.utah.edu*
*Kahlert School of Computing*
*University of Utah, UT, USA*

**John Turnage**                                                        *turnage@math.utah.edu*
*Department of Mathematics*
*University of Utah, UT, USA*

**Zachary Morrow**                                                      *zbmorro@sandia.gov*
*Scientific Machine Learning*
*Sandia National Laboratories*

**John D. Jakeman**                                                     *jdjakem@sandia.gov*
*Optimization and Uncertainty Quantification*
*Sandia National Laboratories*

**Akil Narayan**                                                        *akil@sci.utah.edu*
*Scientific Computing and Imaging (SCI) Institute*
*Department of Mathematics*
*University of Utah, UT, USA*

**Shandian Zhe**                                                        *zhe@cs.utah.edu*
*Kahlert School of Computing*
*University of Utah, UT, USA*

**Varun Shankar**                                                       *shankar@cs.utah.edu*
*Kahlert School of Computing*
*University of Utah, UT, USA*

**Reviewed on OpenReview:** *https://openreview.net/forum?id=Q0C4jYZQ7x*

## Abstract

This paper introduces the Kernel Neural Operator (KNO), a provably convergent operator-learning architecture that utilizes compositions of deep kernel-based integral operators for function-space approximation of operators (maps from functions to functions). The KNO decouples the choice of kernel from the numerical integration scheme (quadrature), thereby naturally allowing for operator learning with explicitly-chosen trainable kernels on irregular geometries. On irregular domains, this allows the KNO to utilize domain-specific quadrature rules. To help ameliorate the curse of dimensionality, we also leverage an efficient dimension-wise factorization algorithm on regular domains. More importantly, the ability to explicitly specify kernels also allows the use of highly expressive, non-stationary, neural anisotropic kernels whose parameters are computed by training neural networks. We present universal approximation theorems showing that both the continuous and fully discretized KNO are universal approximators on operator learning problems. Numerical results demonstrate that on existing benchmarks the training and test accuracy of KNOs is closely comparable to or higher than that of popular neural operators while typically using an order of magnitude fewer trainable parameters, with the more expressive kernels proving important to attaining high accuracy. KNOs thus facilitate low-memory, geometrically-flexible, deep operator learning, while retaining the implementation simplicity and transparency of traditional kernel methods from both scientific computing and machine learning.

# 1 Introduction

Operator learning is a rapidly evolving field that focuses on the approximation of mathematical operators, often those arising from partial differential equations (PDEs). These operators map between infinite-dimensional function spaces and are increasingly employed to reduce the computational cost of simulation-based analyses which require repeated simulations of computationally expensive models. Early approaches to learning PDE surrogates include Khoo et al. (2017), which uses a neural network to map random PDE coefficients to physical quantities of interest and the PDE-Net family Long et al. (2018; 2019), which leverages convolutional filters to learn differential operators that are nonlinearly combined through another network (neural or symbolic) to form the PDE operator. Recent approaches for operator learning include the DeepONet family of neural operators Lu et al. (2021; 2022); Jin et al. (2022), which leverage the universal approximation proof of neural networks for learning nonlinear operators Chen & Chen (1995) and those which hone in on integral transforms.

Motivated by the fact that the solution operators to linear PDEs can be expressed via Green's functions, Graph Neural Operators (GNOs) Li et al. (2020b;a) and the Fourier neural operator family (FNOs) Li et al. (2021); Kovachki et al. (2023); Li et al. (2023; 2024a) form techniques based on composing parameterized integral operators. In particular, the GNO employs a graph-based parametrization to discretize the integral, and the FNO uses a fast Fourier transform (FFT) on an equispaced grid to learn a diagonal matrix-valued kernel in spectral space. Like the FNO, several methods are cognizant of the fact that specific bases diagonalize linear operators and design their architecture around this principle. For instance, Feliu-Fabà et al. (2020); Fan et al. (2019) focuses in on using the non-standard Wavelet transform to learn linear PDEs, while GitNet Wang & Thiery (2023) creates a deep model based on PCA.

Given their success in sequence-to-sequence modeling, transformer-based techniques such as the generalized neural operator transformer (GNOT) Hao et al. (2023) and Transolver Wu et al. (2024) have gained popularity, with Transolver in particular showing that its attention mechanism is an indirect parameterization of an integral operator. Each of these integral-based approaches admits an implicitly parameterized form of the integral kernel, which seats restrictive assumptions and makes it difficult to directly encode kernel properties such as non-stationarity and anisotropy. For instance, in the FNO's discretization kernels are limited to be radial, stationary, and periodic. Moreover, implicit kernel representations contain implicit quadrature rules whose errors are contingent (in unknown ways) on properties of the input function.

In this paper, we propose a novel method, the kernel neural operator (KNO), which extends the FNO family by introducing greater geometric, architectural, and approximation flexibility through the explicit learning of highly expressive, closed-form kernels. The KNO natively supports irregular geometries and scattered data locations by combining interpolation methods with quadrature rules, making it applicable to a wider range of problem domains and aligning the operator with scientific computing workflows that generate the datasets for neural operators in the first place. For problems on regular grids, the KNO allows for a fast dimension-wise factorization of the integral operator to mitigate the curse of dimensionality, ensuring more favorable computational efficiency in high-dimensional settings.

These choices lead to the KNO having a significantly lower memory footprint than other neural operators, an important feature in its own right. In modern operator learning, memory is often the practical bottleneck: recent work on high-resolution neural operators explicitly identifies memory complexity as a barrier to learning at fine resolutions, and newer training methods for neural operators are motivated precisely by the need to reduce optimizer and training-time memory costs Kossaifi et al. (2024); Loeschcke et al. (2025). Many of the most compelling uses of operator surrogates are inherently many-query, including inverse problems, optimization, and related settings in which one must evaluate the surrogate repeatedly rather than only once O'Leary-Roseberry et al. (2024); this further highlights memory complexity and scaling as a relevant issue. Memory characteristics are also important for deployment, where real-time digital-twin and decision-support settings face explicit computational and resource constraints Es-haghi et al. (2024). In this context, a low-memory operator is valuable not only because it is cheaper to store, but because it enables finer discretizations, larger batches, more ensemble evaluations, and operation under tighter hardware budgets. From this perspective, the low-memory profile of the KNO opens up operator-learning regimes that are difficult to access with otherwise accurate but substantially heavier architectures.

Our numerical experiments demonstrate that the KNO achieves comparable or superior accuracy to FNO architectures across a variety of problems. Furthermore, KNO outperforms the more modern transformer-based neural operators in terms of accuracy on several benchmark problems, while requiring **1-2 orders of magnitude fewer trainable parameters** than reported in the literature for FNO, GNOT, and Transolver. This significant reduction in parameter

count results in a much smaller memory footprint, making KNO a more memory-efficient alternative. Importantly, the KNO is both faster to train and infer with than the GNOT and Transolver on even three-dimensional problems, demonstrating that its reduced parameter count, improved accuracy, and ability to generalize to irregular domains does not come at the expense of scalability.

## 1.1 Connections to other methods

Kernel methods have been a cornerstone of machine learning and scientific computing for decades Rasmussen & Williams (2006); Cortes & Vapnik (1995); Boser et al. (1992); Broomhead & Lowe (1988); Sharma & Shankar (2022), with applications ranging from data fitting McCourt et al. (2018); Fasshauer & McCourt (2015) and sparsification Han et al. (2023); Sharma & Shankar (2025) to accelerating physics-informed neural networks Sharma & Shankar (2022) and enhancing DeepONets Sharma & Shankar (2025). They have also long played a central role in scientific computing for integral operators Gingold & Monaghan (1977); Peskin (2002); Kassen et al. (2022a;b); Hsiao & Wendland (2008); Cortez (2001); Shankar & Olson (2015) and finite-difference-type discretizations Wright & Fornberg (2006); Fornberg & Flyer (2015); Bayona et al. (2019); Fasshauer & McCourt (2015); Shankar et al. (2014); Shankar & Fogelson (2018). In this sense, the KNO should be viewed not as an isolated neural-operator architecture, but as part of a broader and older line of work in which continuous kernels, quadrature, and structured approximation are used to parameterize nonlocal mappings between function spaces.

Important antecedents and neighboring approaches include early neural surrogates for parametric PDE maps Khoo et al. (2017), wavelet/compressed-operator architectures such as BCR-Net Fan et al. (2019) and pseudo-differential meta-learning Feliu-Fabà et al. (2020), PDE-Net and PDE-Net 2.0 Long et al. (2018; 2019), and physics-informed FNO variants such as PINO Li et al. (2024b). This perspective also places the KNO within a wider prehistory of modern operator learning. Prior to the recent DeepONet/FNO era, several works already learned parametric PDE maps or compressed operator families using architectures motivated by numerical analysis. For example, Khoo, Lu, and Ying Khoo et al. (2017) learned parametric PDE solution maps directly from coefficient fields, while BCR-Net Fan et al. (2019) and the subsequent pseudo-differential meta-learning framework of Feliu-Fabà, Fan, and Ying Feliu-Fabà et al. (2020) exploited multiresolution and wavelet-compression ideas to represent operator families efficiently. PDE-Net and PDE-Net 2.0 Long et al. (2018; 2019) are also important antecedents, although their primary goal is PDE identification and dynamics prediction from trajectory data rather than the discretization-transfer operator-learning setting considered here. These works help clarify that many of the central themes in current neural-operator research—compressed representations, structure-aware parameterizations, and learning maps induced by differential or integral operators—have deep roots in scientific machine learning and numerical analysis.

Recent operator-learning architectures with related efficiency or discretization-flexibility claims include interpolation-based kernel operators Batlle et al. (2024), BelNet Zhang et al. (2023), GIT-Net Wang & Thiery (2023), ensemble and mixture-of-experts DeepONets Sharma & Shankar (2025), and, more conceptually, Integral Neural Networks Solodskikh et al. (2023). Interpolation-based kernel operator learning methods Batlle et al. (2024) already demonstrate that classical kernel machinery can be adapted to operator learning with very few trainable parameters, albeit with lower accuracy than the KNO in most of our experiments. BelNet Zhang et al. (2023) is a mesh-free neural operator that learns a latent (shallow) neural basis using a kernel integral operator as a starting point and reconstructs the output on potentially different input and output meshes.

Among recent competitors, GIT-Net Wang & Thiery (2023) is perhaps the closest to the KNO in overall spirit, since it is also motivated by integral representations of PDE operators. The distinction, however, is substantial. GIT-Net parameterizes adaptive generalized integral transforms in specialized functional bases and derives its efficiency from the existence of a parsimonious transform representation Wang & Thiery (2023). The KNO, in contrast, does not rely on learning such a global transform. It learns the kernel of the operator directly and applies quadrature to realize the operator action. In this sense, GIT-Net is a transform-learning approach, whereas the KNO is a direct kernel-learning approach. This difference is important on irregular domains and for heterogeneous discretizations, where kernel locality and quadrature can be easier to deploy than a global transform parameterization.

Ensemble and mixture-of-experts variants of DeepONet Sharma & Shankar (2025) similarly seek greater expressivity and efficiency through basis enrichment, locality, and sparsity; the KNO outperforms this method as well. At a broader architectural level, Integral Neural Networks Solodskikh et al. (2023) are also relevant conceptually, since they replace discrete layer tensors with continuous integral parameterizations and use quadrature to realize networks at variable discretizations, although they were not introduced specifically as PDE neural operators.

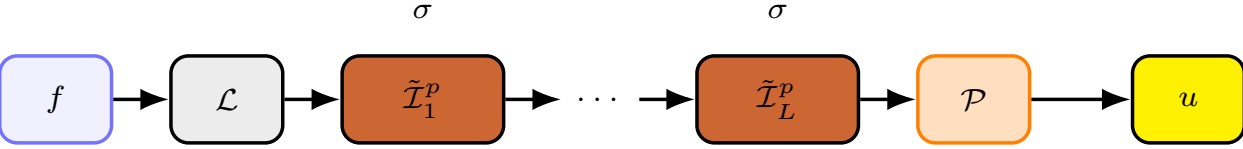

Figure 1: KNO: the input function $f$ is lifted, passed through activated kernel integral layers $\tilde{\mathcal{I}}^p$ (discretized by quadrature and augmented by pointwise channel mixing; see (14)), and projected by $\mathcal{P}$ to the output function $u$. Here, $\sigma$ denotes the pointwise activation function.

Against this backdrop, the KNO occupies a distinct point in the design space. Like several of the methods above, it leverages an integral viewpoint, but it does so through trainable matrix-valued kernel layers whose scalar kernel components are subsequently discretized by quadrature. This gives the KNO a direct connection to classical kernel approximation, Nyström-type quadrature ideas, and kernel-based scientific computing, while still retaining the compositional expressivity of deep operator learning. In contrast to architectures that rely on global spectral grids, learned coordinate deformations, or large latent bases, the KNO obtains nonlocality, discretization flexibility, and parameter efficiency directly from the kernel representation itself. In particular, the scalar-valued kernels in the KNO offer an explicit mechanism for non-stationary and anisotropic approximation, while the matrix-valued kernel structure provides a principled way to couple channels without requiring very large dense parameterizations.

This distinction is especially important on irregular domains. DeepONet-type models Lu et al. (2022); Peyvan et al. (2024) can accommodate irregular geometries, but they typically require all input functions to be observed at a common set of sensor locations. The FNO has been extended beyond regular Cartesian grids through architectures such as GeoFNO and related methods Li et al. (2023; 2024a), but these approaches still rely on Fourier machinery together with learned mappings or geometric embeddings into a representation amenable to FFT-based convolution. Transformer-based neural operators Hao et al. (2023); Wu et al. (2024) and BelNet Zhang et al. (2023) can also work on irregular domains, but typically at the expense of larger learned representations. By contrast, the KNO requires only evaluation or interpolation of functions at quadrature sites, after which the operator action is realized through the kernel itself. This makes the approach naturally compatible with irregular domains, unstructured samplings, and settings in which different samples are observed on different discretizations.

Finally, physics-informed operator learning, such as PINO Li et al. (2024b), is best viewed as complementary to the KNO rather than as a competing architectural philosophy. PINO augments an FNO-type operator learner with PDE-residual and physics-based constraints during training. The analogous idea could in principle be incorporated into the KNO as well, by supplementing data mismatch with residual, boundary, or variational losses evaluated on the learned kernelized operator outputs. We therefore view physics-informed training as an orthogonal axis to the present contribution: the architectural novelty of the KNO lies in its kernelized integral parameterization, while physics-informed objectives represent a potentially valuable extension that could be combined with this architecture in future work.

## 2 Kernel Neural Operators (KNOs)

Given Euclidean domains $\Omega_u, \Omega_y$ and $d_u, d_y \in \mathbb{N}$, neural operators learn mappings from a Banach space $\mathcal{U} = \left(\Omega_u; \mathbb{R}^{d_u}\right)$ of $\mathbb{R}^{d_u}$-valued functions to a Banach space $\mathcal{Y} = \mathcal{Y}(\Omega_y; \mathbb{R}^{d_y})$ of $\mathbb{R}^{d_y}$- valued functions through supervised training on a finite number of input-output measurements. From a statistical learning point of view, neural operators are learned from measurements of input functions drawn from a probability measure $\nu$ on $\mathcal{U}\left(\Omega_u; \mathbb{R}^{d_u}\right)$. In the following, we present the formulation of KNOs, which are a special class of neural operators that leverage properties of certain kernel functions for the benefit of efficiency and accuracy; a schematic is shown in Figure 1.

### 2.1 Function Space Formulation

**Architecture** Let $\mathcal{G}$ be an unknown operator we wish to learn that is an element of the $L^2$-type Bochner space $L^2_\nu(\mathcal{U}; \mathcal{Y})$, i.e., $\mathcal{G}$ is a mapping from $\mathcal{U}$ to $\mathcal{Y}$ that is Borel-measurable with respect to the probability measure $\nu$ on $\mathcal{U}$. We are interested in learning a KNO $\mathcal{G}^\dagger$ that minimizes a loss function $L$ measuring how well functions predicted by

the operator match the training data. For example, the loss function may be the $L^2_\nu$ norm on operators,

$$L(\mathcal{H}, \mathcal{G}) = \|\mathcal{H} - \mathcal{G}\|^2_{L^2_\nu(\mathcal{U};\mathcal{Y})} = \mathbb{E}_{f\sim\nu}\|\mathcal{H}(f) - \mathcal{G}(f)\|^2_\mathcal{Y},$$

which is the loss function we use in our experiments, with the addition of some regularization on the kernel scale parameters and a scaling term to account for relative error. The corresponding statistical learning problem is

$$\mathcal{G}^\dagger = \underset{\mathcal{H}\in\text{KNOs}}{\arg\min}\, L(\mathcal{H}, \mathcal{G}), \tag{1}$$

where KNOs are operators of the form

$$\mathcal{H} = \mathcal{P} \circ \sigma \circ \mathcal{I}^p_L \circ \sigma \circ \mathcal{I}^p_{L-1} \circ \sigma \circ \ldots \sigma \circ \mathcal{I}^p_1 \circ \mathcal{L}. \tag{2}$$

The operators $\mathcal{I}_\ell, \mathcal{L}, \mathcal{P}$ are all trainable, and an appropriate parameterization of these defines a KNO, and $p \in \mathbb{N}^+$ is a hyperparameter that defines a *channel* dimension. The function $\sigma$ is a nonlinear activation that operates pointwise: $(\sigma \cdot f)(x) := \sigma(f(x))$; we used GeLU Hendrycks & Gimpel (2016). Additionally, the initial operator $\mathcal{L}$ is a *lifting operator* that takes $\mathbb{R}^{d_u}$-valued functions to $\mathbb{R}^p$-valued functions, i.e., creates $p$ channels. The ultimate operator $\mathcal{P}$ is a *projection operator* that takes $\mathbb{R}^p$-valued functions and compresses them down to $\mathbb{R}^{d_y}$-valued functions.

**Integral operators**  The integral operators $\mathcal{I}^p_\ell$ are linear operator mappings from vector-valued functions to vector-valued functions. These operators are defined by,

$$\mathcal{I}^p_\ell(\boldsymbol{f}_\ell) = \int_{\Omega_y} \boldsymbol{K}^{(\ell)}(x, y)\boldsymbol{f}_\ell(y)dy, \tag{3}$$

$$\boldsymbol{f}_\ell : \Omega_y \to \mathbb{R}^p, \ \boldsymbol{g}_\ell = \mathcal{I}^p_\ell(\boldsymbol{f}) : \Omega_y \to \mathbb{R}^p, \tag{4}$$

where $\boldsymbol{K}^{(\ell)} : \Omega_y \times \Omega_y \to \mathbb{R}^{p\times p}$ is a matrix-valued kernel function,

$$\boldsymbol{K}^{(\ell)}(x, y) = \begin{pmatrix} K^{(\ell)}_{1,1}(x, y) & \ldots & K^{(\ell)}_{1,p_{\ell-1}}(x, y) \\ K^{(\ell)}_{2,1}(x, y) & \ldots & K^{(\ell)}_{2,p_{\ell-1}}(x, y) \\ \vdots & \ddots & \vdots \\ K^{(\ell)}_{p_\ell,1}(x, y) & \ldots & K^{(\ell)}_{p_\ell,p_{\ell-1}}(x, y) \end{pmatrix}. \tag{5}$$

The overall structure closely resembles FNOs, but differs in an important aspect: FNOs implicitly impose a diagonal structure on this matrix-valued kernel (as we also do), but further force that the individual scalar-valued kernels be isotropic and stationary due to the parametrization of the integrals operators $\mathcal{I}_\ell$ via the FFT. In contrast, the KNO *decouples the discretization of the integral operators $\mathcal{I}^p_\ell$ from the choice of kernel, thereby allowing the use of very general and highly-expressive kernels.* The actual integration is accomplished through multi-dimensional quadrature in the general case, though we also leverage a special dimension-wise factorization algorithm on regular grids that removes the need for multi-dimensional quadrature. The strength of the KNO lies in this decoupling: the ability to freely choose kernels allows us to choose highly expressive kernels with a very small number of trainable parameters (in comparison to other neural operators), while the ability to freely select quadrature locations allows us to tackle arbitrary domains (while also efficiently tackling regular ones). We now describe the KNO in further detail; mathematical formulations are shown in (2) and (14).

**Remarks**  As in many neural operator formulations, we augment our kernel integral operators (3) at the discrete level with dense cross-channel affine transformations ("pointwise convolutions") having trainable parameters; we describe these in Appendix A.1.1.

## 2.2 Discretization

We now describe the ingredients required to discretize the KNO. Once lifting and projection operators are chosen, the KNO requires a set of concrete choices: (1) a matrix-valued kernel in the lift-dimension (Section 2.2.1); (2) scalar-valued kernels for use within the matrix-valued kernel (also Section 2.2.1); (3) an interpolant to allow for data transfer from training locations to quadrature locations along with a training objective (Section 2.2.2); (4) a problem-dependent quadrature rule (Section 2.2.3) This is visualized for a single KNO layer in Figure 2. The fully discretized KNO is shown in Section 2.2.4.

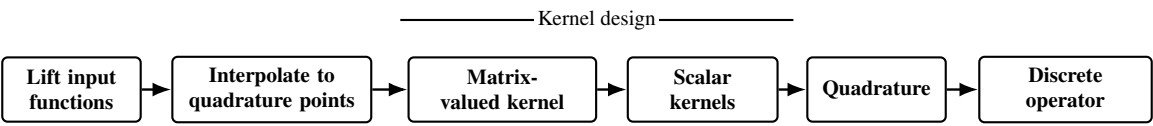

Figure 2: Discretization pipeline for one KNO layer. The input is first lifted to a latent channel representation and then interpolated onto the quadrature grid. Kernel design proceeds in two stages: one first chooses a matrix-valued kernel governing interactions across channels, and then specifies the underlying scalar spatial kernels. Quadrature is finally applied to discretize the integral operator.

### 2.2.1 Kernel design

The KNO requires kernel design to be done at two levels. First, a choice must be made on the structure of the matrix-valued kernel (MVK) defined over the channels, then the individual scalar-valued kernels within the MVK must be chosen. We chose a diagonal MVK in this work by setting

$$\left(\boldsymbol{K}^{(\ell)}(x,y)\right)_{ij} = \begin{cases} K_{i,j}^{\ell}(x,y), & \text{if } i = j \\ 0, & \text{otherwise} \end{cases}, \tag{6}$$

where $i, j = 1, \ldots, p$. This choice aligns with findings on the inherent low-rank nature of the matrices associated with integral operators Fan et al. (2019) and also with a similar choice made in the FNO architecture. However, the KNO allows for other choices also. For instance, we found other MVKs with greater fill-in to be beneficial when using single-parameter scalar-valued kernels and a small channel dimension $p$, but concluded that the diagonal MVK was the easiest to train and the most robust across problems when a sufficiently-expressive scalar-valued kernel was used; for experimental evidence, see Appendix A.7.2. As in the FNO, we use pointwise convolutions to ensure that information mixes across the $p$ channels, but we found that removing these convolutions only resulted in a mild degradation of accuracy; see Appendix A.7.4. Next, the individual scalar-valued kernel entries in the matrix-valued kernel must be chosen. After exploring a variety of kernels (including compactly-supported kernels, purely radial kernels, and many others), we settled on two highly expressive kernels. The best-performing across kernel across most problems was a *neural*, non-stationary, anisotropic, generalized spectral mixture kernel (NS-GSM) given by Remes et al. (2017)

$$K_{\text{NS-GSM}}^{\ell}(x,y) = \sum_{i=1}^{Q} w_i(x)w_i(y)k_{\text{Gibbs},i}(x,y)g_i(x,y), \tag{7}$$

where $\mu_1(x), \ldots, \mu_Q(x)$ are each vector-valued latent frequency functions; $w_1(x), \ldots, w_Q(x)$ are scalar-valued latent amplitude functions; and $g_i(x,y) = \cos\left(2\pi\left(\mu_i(x)^{\top}x - \mu_i(y)^{\top}y\right)\right)$. Here, $k_{\text{Gibbs},i}(x,y)$ is the Gibbs kernel (itself a non-stationary generalization of the Gaussian kernel Gibbs (1997); Heinonen et al. (2016); Paciorek & Schervish (2004)) given by

$$k_{\text{Gibbs},i}(x,y) = \sqrt{\frac{2s_i(x)s_i(y)}{r_i(x,y)}}\exp\left(\frac{-(x-y)^2}{r_i(x,y)}\right),$$

where $s_i(x)$ is a latent length-scale function that is the $i$-th component of a vector-valued length scale function $\mathbf{s}(x) = [s_1(x), \ldots, s_Q(x)]$, and $r_i(x,y) = s_i(x)^2 + s_i(y)^2$. The latent length-scale, frequency, and amplitude functions are each obtained as the outputs of a single shallow feedforward neural network of the form NN : $\mathbb{R}^{d_y} \to \mathbb{R}^{2Q+Qd_y}$ with a single hidden layer; we used a SELU activation on the hidden layer Klambauer et al. (2017) and a softplus activation on the output layer Dugas et al. (2000), the latter ensuring that the latent quantities are always positive. We also explored use of the following stationary Gaussian spectral mixture (GSM) kernel, which simply omits the non-stationary Gibbs kernel and uses trainable weights $w_i$, frequencies $\mu_i$, and diagonal covariances $\Sigma_i$ (not output by neural networks):

$$K_{\text{GSM}}^{\ell}(x,y) = \sum_{i=1}^{Q} w_i \exp\left(-\tfrac{1}{2}(x-y)^{\top}\Sigma_i(x-y)\right)\cos\left(2\pi\mu_i^{\top}(x-y)\right). \tag{8}$$

In general, we found that the NS-GSM kernel outperformed the GSM kernel on all but two test problems: these test problems involved noisy datasets (due to real-world measurements and fluid turbulence). This is likely due to the difficulty of training the more expressive NS-GSM kernels in the presence of noise in the dataset. This is also discussed in Section 4. We also present further ablations over kernel types in Appendix A.7.1.

**Dimension-Wise Factorization**    The KNO is able to use multidimensional quadrature in all practical settings. However, for problems with data lying on regular grids, it is possible to factorize the integral operator in (3) to obtain large speedups and combat the curse-of-dimensionality; as a by-product, this allows us to use univariate quadrature rules also. Letting $\Omega_y = [a, b]^d$ (without loss of generality), $x = (x_1, \ldots, x_d)$, and $y = (y_1, \ldots, y_d)$, we write:

$$\mathcal{I}_\ell^p(\boldsymbol{f})(x) = \sum_{j=1}^d \int_a^b \boldsymbol{K}_j^{(\ell)}(x_j, y_j) \boldsymbol{f}(x_1, \ldots, x_{j-1}, y_j, x_{j+1}, \ldots, x_d) dy_j, \tag{9}$$

where $\boldsymbol{K}_j^\ell$ are MVKs chosen for each coordinate direction. In practice, since we use diagonal MVKs, one only needs to choose coordinate-wise scalar-valued kernels in the KNO. Regardless, this now requires only the use of univariate quadrature. Note that this technique does not require our *kernels* themselves to be factorizeable. Interestingly, this dimension-wise factorization allows the KNO to have better asymptotic computational complexity properties than the FFT when considering dimensional scaling, since the cost now only linearly increases with dimension. Consequently, we do not use the FFT in this work.

### 2.2.2   Sampling and outer discretization

Numerically constructing (2) requires sampling from $\nu$ and a discretization of $\|\cdot\|_\mathcal{Y}$. To this end, we trained our KNOs using $M$ independent and identically distributed input samples of functions $f^{(m)} \sim \nu$ drawn from $\mathcal{U}$ and the associated output function data $g^{(m)} := \mathcal{G}(f^{(m)})$, for $m \in [M]$. We used a *set of training points (locations)*, $X_T = \{x_j\}_{j\in[N_T]} \subset \Omega$, to both discretize the input and output functions $f^{(m)}$ and $g^{(m)}$ and to approximate the norm $\|\cdot\|_\mathcal{Y}$. Hence, during learning we optimized

$$\|\mathcal{H} - \mathcal{G}\|_{L^2_\mu(\mathcal{U}, \mathcal{Y})}^2 \stackrel{f^{(m)} \sim \nu}{\simeq} \frac{1}{MN_T} \sum_{(m,j)\in[M]\times[N_T]} \left\| \mathcal{H}(f_{X_T}^{(m)})(x_j) - g^{(m)}(x_j) \right\|_2^2, \tag{10}$$

which states that the $L^2_\mu$ norm of the operator on the left is approximated by the discretization on the right, with the functions $f^{(m)}$ drawn from the probability distribution with measure $\nu$. Note that once the KNO is trained, it can be evaluated at any locations in the domain of the input function, *i.e.,* training and evaluation grids need not coincide.

Since the KNO potentially decouples the training grid from the integral operators, we now have two distinct cases to tackle.

**Irregular Domains**    In the most general case (on irregular domains), the training grid $X_T$ is typically distinct from the quadrature points used for numerical integration (to be introduced shortly). In this case, we first employ the channel lift $\mathcal{L}$ which produces samples of $\mathbb{R}^p$-valued functions on the training grid, then use a trainable kernel interpolant to transfer the lifted function $\mathbf{f}$ to the quadrature points:

$$\boldsymbol{f}_{X_T}(x) \approx \sum_{n\in[N_T]} K(x, x_n)\boldsymbol{c}_n, \tag{11}$$

where the $\boldsymbol{c}_n$ are determined through a size-$N_T$ linear system solve that enforces $\boldsymbol{f}_{X_T}(x_n) = \boldsymbol{f}(x_n)$; this particular system has $p$ right hand sides. We also explored using the interpolation *before* the channel lift $\mathcal{L}$. While both choices performed well, we found that using the interpolation after the lift operator reduced the need for pointwise convolutions in the integration layers since the interpolant itself produces coupling between the layers; see Appendix A.7.4 for ablations. Further, we found that interpolating after the channel lift alleviated the Runge phenomenon, which is seen when interpolating infintely-smooth target functions sampled on grids of equispaced points Platte et al. (2011); this is likely because the lifted input functions are not as smooth as the input functions themselves.

Regardless, this interpolant can be viewed as part of the lifting operator $\mathcal{L}$ and allows for evaluation of $\boldsymbol{f}$ at the quadrature points to be introduced shortly. We also leverage a separate kernel interpolant similar to the one in (11) to transfer information from the quadrature points *back* to the training points to evaluate the objective function in (10); this second interpolant can be viewed as part of the projection operator $\mathcal{P}$. For these interpolants, we ablated over a variety of kernels and used problem-dependent kernels (selected through ablation). We discuss these in Section 4.

**Regular Grids**    When the training points form a regular (tensor-product) grid, we exploit the tensor-product structure to perform the dimension-wise factorization of the kernel integral operator as in (9) in conjunction with simple univariate quadrature rules (discussed shortly). Consequently, in this scenario, the training points also serve as quadrature points and kernel interpolants are not needed for data transfer.

### 2.2.3 Integral operator discretization: Quadrature

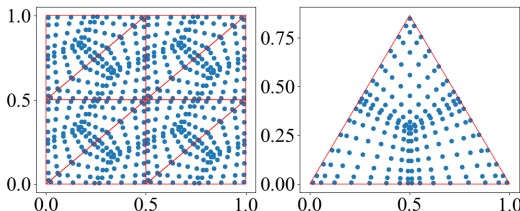

Figure 3: Clustered quadrature points on $[0,1]^2$ (left) and a reference triangle (right).

In order to propagate $f_{X_T}$ through $\mathcal{H}$ in (10), one must discretize all the integral operators; we accomplished this with quadrature. Consider the discretization of an integral operator $\int_\Omega K(x,y)f(y)d\mu(y)$ that acts on a scalar-valued function $f : \mathbb{R}^d \to \mathbb{R}$; the generalization to vector-valued functions is straightforward. Then given a *quadrature rule* $\{w_i^q, y_i^q\}_{i=1}^{N_Q}$, where $w_i^q \in \mathbb{R}$ are *quadrature weights* and $y_i^q \in \mathbb{R}^d$ are *quadrature points*, the quadrature-based discretization of a KNO integral operator is

$$\int_\Omega K(x,y)f(y)d\mu(y) \approx \sum_{i=1}^{N_Q} w_i^q K\left(x, y_i^q\right) f(y_i^q). \tag{12}$$

We tailor the choice of quadrature rule to the domain of the problem, and thus employ several quadrature rules in this work, discussed below.

**Irregular Domains** On 2D irregular domains $\Omega$, we tessellated $\Omega$ with a triangle mesh that divided $\Omega$ into some set of nonoverlapping triangles $\Omega_\ell$, $\ell = 1, \ldots, N_\Omega$ such that

$$\int_\Omega K(x,y)f(y)d\mu(y) = \sum_{\ell=1}^{N_\Omega} \int_{\Omega_\ell} K(x,y)f(y)d\mu(y). \tag{13}$$

Following standard scientific computing practices Karniadakis & Sherwin (2005); Cantwell et al. (2015) we discretized (13) using a quadrature rule for each of the subdomains $\Omega_\ell$ affinely-mapped from a symmetric quadrature rule on a standard ("reference") simplex $\Omega_{ref}$ in $\mathbb{R}^d$ Freno et al. (2020); see Figure 3. In Section A.4.2, we also present results on a 3D problem within the unit ball that utilized a quadrature rule specially tailored for that domain von Winckel (2025). In general, one can use any reasonable quadrature rule in the KNO. In general, the use of quadrature introduces the curse of dimensionality into the discretization of the KNO; for general domains, this can be potentially ameliorated with sparse grids Holtz (2011) or Monte Carlo Dick (2016) techniques. We further discuss the computational complexity of quadrature in Appendix A.1.4.

**Cartesian Grids** On Cartesian grids, we used the dimension-wise factorized kernel integral operator (9), and thus only required univariate quadrature rules. We found that the (composite) univariate trapezoidal rule was sufficiently accurate Davis & Rabinowitz (1984); Quarteroni et al. (2000); Atkinson (1989); Stoer & Bulirsch (2002); this rule converges as $O(h^2)$ for general functions (where $h$ is the grid spacing) and exponentially for periodic functions Trefethen & Weideman (2014)[1]. This discretization is highly efficient and avoids the curse of dimensionality as it allows for $O(\mathbb{R}^{d_y})$ sums of $N_Q$ terms each rather than one $O(N_Q^{d_y})$ sum.

### 2.2.4 Discretized KNO

In summary, the discretized KNO $\widetilde{\mathcal{H}}$ that we used to numerically construct $\mathcal{H}$ in (2) can be written as a function that takes in $f_{X_T}$ and returns an approximation to the output function $\mathcal{H}(f)$ evaluated at $X_T$:

$$\widetilde{\mathcal{H}}(f_{X_T}) = \left(\mathcal{P} \circ \sigma \circ \tilde{\mathcal{I}}_L^p \circ \ldots \sigma \circ \tilde{\mathcal{I}}_1^p \circ \mathcal{L}\right)(f_{X_T}) \tag{14}$$

---

[1]If necessary, one can always use the KNO with a higher-order accurate quadrature rule, but we found that quadrature errors were generally smaller than training errors.

where $\left(\tilde{\mathcal{I}}_k^p\right)$ are now the discretized integral operators incorporating pointwise convolutions, $\mathcal{L}$ is a discretized lifting operator (potentially incorporating an interpolant of the form (11)), and $\mathcal{P}$ is a discretized projection operator (potentially also incorporating an interpolant); details on the neural network architectures used in $\mathcal{L}$ and $\mathcal{P}$ are presented in Appendix A.1.2. Much like in the FNO, the discretized integral operators also include a pointwise convolution that aggregates information across channels; this is discussed in Appendix A.1.1.

## 3 Universal Approximation Theorems

We now present two universal approximation theorems for the KNO, with the goal of establishing that the KNO is a safe and reasonable choice for operator learning; we defer their proofs to Appendix B. We defer theorems about convergence rates and sampling to future work. The first theorem is a universal approximation theorem for the infinite-dimensional KNO (2).

**Theorem 3.1.** *Let $\Omega \subset \mathbb{R}^d$ be compact, and let $A \subset (L^2(\Omega;\mathbb{R}), \|\cdot\|_{L^2(\Omega)})$ be compact. Let $\mathcal{G} : A \to (L^2(\Omega;\mathbb{R}), \|\cdot\|_{L^2(\Omega)})$ be a continuous operator. For any $\epsilon > 0$, there exists a KNO $\mathcal{H} : A \to L^2(\Omega;\mathbb{R})$ of the form (2) with continuous positive-definite kernels $K_{i_\ell,j_\ell}^{(\ell)}$ such that*

$$\sup_{f \in A} \|\mathcal{H}[f] - \mathcal{G}[f]\|_{L^2(\Omega)} < \epsilon. \tag{15}$$

*Proof.* The proof is given in Appendix B.1. $\qquad\square$

The second theorem shows that the fully discretized KNO (14) can recover the infinite-dimensional version to arbitrary accuracy.

**Theorem 3.2.** *Adopt the same assumptions as Theorem 3.1, but with $A' \subset C^1(\Omega;\mathbb{R})$, compact with respect to the $\|\cdot\|_{L^\infty}$ norm and with uniformly bounded first derivatives. Additionally, let $\{w^{(M)}\}_{M \in \mathbb{N}}$ and $\{x^{(M)}\}_{M \in \mathbb{N}}$ define a sequence of $M$-point quadrature rules on $\Omega$. Suppose that there exists $C > 0$ such that, for any $f \in C^1(\Omega;\mathbb{R})$,*

$$\left| \sum_{m \in [M]} w_m^{(M)} f(x_m^{(M)}) - \int_\Omega f(x)\,dx \right| \leq \frac{C\|\nabla f\|_{L^\infty}}{M}.$$

*For any $\epsilon > 0$, there exists $M \in \mathbb{N}$, $\nu > 0$, and $\widetilde{\mathcal{H}}_M : \mathbb{R}^{N_T} \to \mathbb{R}^{N_T}$ of the form (14) such that*

$$\sup_{f \in A'} \left\| \widetilde{\mathcal{H}}_M(\boldsymbol{f}_T) - (\mathcal{G}[f])\big|_{X_T} \right\|_{\ell^\infty(\mathbb{R}^M)} < \epsilon + \nu\, h_{\Omega,X_T}, \tag{16}$$

*where $\boldsymbol{f}_T = \{f(x)\}_{x \in X_T}$,*

$$h_{\Omega,X_T} = \sup_{x \in \Omega} \min_{x_i \in X_T} \|x - x_i\|$$

*is the fill distance, and $\widetilde{\mathcal{H}}_M$ depends parametrically on the quadrature nodes $X_M = \{x_m^{(M)}\}_{m \in [M]}$.*

*Proof.* The proof is given in Appendix B.2. $\qquad\square$

## 4 Results

We now describe our numerical experiments with KNOs and other state-of-the-art neural operators on seven different benchmark problems from literature. We present results on both tensor-product domains (all of which used boundary-anchored equidistant grids) and irregular domains (which used regular grids, triangle meshes, or point clouds). The KNO models were all trained using the Adam optimizer Kingma & Ba (2015) with a cyclic cosine annealing learning rate schedule. All models were trained for $10,000$ epochs on all benchmark problems; the exception was on the NS-Pipe example (below), where we trained the KNO for 500 epochs (to match the reported GeoFNO, GNOT, and Transolver results). Other technical details are described in Appendix A.1. We measured the accuracy of our KNOs by computing the mean and standard error of the $\ell_2$ relative errors (on generalization) of each KNO obtained from four different training runs with different random model parameter initializations. These errors were compared to those of the FNO Li et al. (2021), the GNOT Hao et al. (2023), the Transolver Wu et al. (2024), and KM Batlle et al.

(2024); Appendix A.2.2 discusses these other models in greater detail. We used publicly available code for the FNO, the GNOT, and the Transolver to generate all results except those for the NS-Pipe example; for this latter example, we report the GNOT and Transolver results from Wu et al. (2024) and the (Geo-)FNO result from Li et al. (2023). For the two problems on irregular domains – Darcy (triangle) and Reaction-Diffusion – we report results from the Geo-FNO Li et al. (2023) in the FNO column, since the FNO cannot be directly applied to these domains. We normalized the training inputs to have mean zero and unit standard deviation, and re-scaled the predicted functions based on these inputs to match the scaling of the ground truth output functions in all cases. Below, we briefly describe the problems

Table 1: Problem geometries and data sampling locations.

| **Problem** | Geometry | Sample Locs. |
| --- | --- | --- |
| Burgers' | Unit interval | Regular grid |
| Beijing-Air | Unit interval | Regular grid |
| Darcy (PWC)) | Unit square | Regular grid |
| Darcy (triangle) | Triangle | Triangle mesh |
| NS-Pipe | Cubic spline (curved) | Regular grid |
| NS Mach 1.0 | Unit cube | Regular grid |
| React-Diff. | Unit ball | Point Cloud |

that we compared the different methods on. These problems are described in greater detail in Appendices A.3 and A.4. In Table 1, we summarize the type of the problem geometry and the data sample locations.

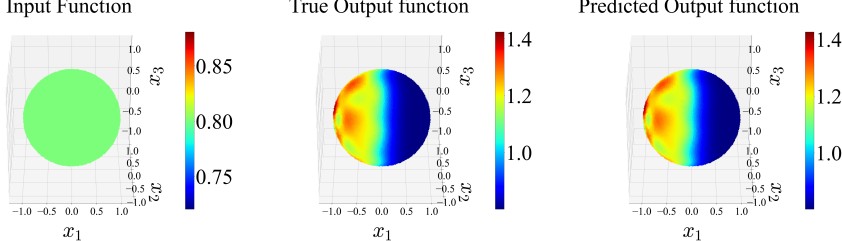

Figure 4: The 3D reaction-diffusion problem A.4.2, where an input function is given (left), the true output function (center), and a prediction from the KNO (right).

**1D Burgers' Equation**: Predict the solution $u_1 : (0, 1) \to \mathbb{R}$ of the one-dimensional viscous Burgers' equation at time $t = 1$, given the initial condition $u_0 : (0, 1) \to \mathbb{R}$, with the viscosity set to $\nu = 0.1$.

**1D Beijing-Air Problem**: Predict the hourly concentration of CO over the following week based on the previous week's measurements of $SO_2$, CO, PM2.5, and PM10, using the Beijing-Air[2] dataset. This dataset contains hourly measurements of several air pollutants in Beijing collected between 2014 and 2017. For this task, 5,000 weeks were randomly selected for training and 1,000 weeks for testing. The KNO results are visualized in Appendix A.3.2.

**2D Darcy Flow (PWC)**: Predict the pressure field $u : [0, 1]^2 \to \mathbb{R}$ from the given piecewise constant (PWC) permeability field $c : [0, 1]^2 \to \mathbb{R}$, based on the Darcy flow equation Lu et al. (2022). The permeability fields are sampled from Gaussian random fields and thresholded to create PWC functions.

**2D Darcy Flow on a Triangular Domain**: Predict the pressure field $h(x, y)$, computed using Darcy's equation, from the given boundary condition on a triangular domain Lu et al. (2022), with the permeability set to $0.1$ and the forcing set to $-1$. The boundary condition fields are sampled from Gaussian random fields, and the data lies on an 861-node uniform triangular mesh.

**2D Incompressible Navier–Stokes Equation in a Pipe**: Predict the final velocity field for flow in a 2D pipe governed by the incompressible Navier–Stokes equations with viscosity $\nu = 0.005$, based on the benchmark problem from Li et al. (2023); the input function is the pipe geometry itself. A parabolic inlet profile $\mathbf{v} = [1, 0]$ is imposed, with a free

---

[2]https://archive.ics.uci.edu/dataset/501/beijing+multi+site+air+quality+data

outflow boundary condition at the outlet and no-slip walls. The pipe (length 10, width 1) follows a centerline defined by a piecewise cubic polynomial formed by the vertical positions and slopes at five spatially uniform control nodes. While the domain is irregular, the data is provided on a structured mesh. The final velocity field is taken from the dataset in Li et al. (2023), though the final time is not reported in that work.

**3D Compressible Navier–Stokes (NS) Equations in a Torus**: Predict the velocity field $v_1 : [0,1]^3 \to \mathbb{R}$ after one time step from an initial random velocity field $v_0 : [0,1]^3 \to \mathbb{R}$, based on the compressible Navier–Stokes equations in a challenging setting with a Mach number of 1.0. Periodic boundary conditions create a toroidal geometry, as described in Takamoto et al. (2022).

**3D Reaction-Variable-Coefficient-Diffusion on a Point Cloud**: Predict the chemical concentration $c(y, t = 0.5) : \mathbb{R}^3 \to \mathbb{R}$ at $t = 0.5$ from uniform initial concentrations $c(y, t = 0) : \mathbb{R}^3 \to \mathbb{R}$, based on the reaction-diffusion equation with a spatially varying diffusion coefficient and discontinuous reaction rates. The problem is solved within the interior of the unit ball, where the concentrations at the final time exhibit sharp spatial gradients. The data is sampled on a point cloud inside the unit ball, as described in Sharma & Shankar (2025). KNO predictions are visualized in Figure 4.

## 4.1 Relative errors

Table 2: Percent $\ell_2$ Relative Errors on Generalization. The table reports the errors for the best-performing KNO, FNO, GNOT, KM, and Transolver operators. Standard errors are provided in Section A.5. The entry "–" indicates that the Transolver can not make predictions when the output and input functions have heterogeneous grids, as is the case in the Darcy (triangle) problem (this was not a focus of their implementation).

| Problem | FNO | GNOT | Transolver | KM | KNO |
|---|---|---|---|---|---|
| Burgers' | **0.276** | 0.89 | 1.077 | 2.831 | 0.574 |
| Beijing-Air | 55.33 | 40.3 | 26.273 | 50.982 | **24.941** |
| Darcy (PWC) | 1.79 | 2.58 | 1.99 | 3.06 | **1.55** |
| Darcy (triangle) | 0.043 | 0.111 | – | **0.033** | 0.045 |
| NS-Pipe | 0.67 | 0.47 | 0.33 | 2.742 | **0.317** |
| NS Mach 1.0 | 58.05 | 81.5 | **48.127** | 54.150 | 48.636 |
| React.-Diff. | 6.68e-3 | 4.47e-3 | 8.09e-3 | **8.75e-05** | 9.20e-4 |

We evaluated the performance of the KNO on the aforementioned benchmarks, which span varied geometries, dimensionalities, and physical systems. The results, presented in Table 2, highlight the KNO's ability to achieve high accuracy across all tasks, demonstrating its effectiveness in modeling complex operator mappings. Additionally, Table 3 provides details on the kernel and quadrature choices used for each problem, showcasing the adaptability of the KNO's architecture to different computational requirements and domains.

Before presenting the results, we note that for problems on irregular grids (Table 1), we used an anisotropic Gaussian kernel for interpolation between the training grid and quadrature points, defined by a trainable Mahalanobis distance $(x - y)^T L L^T (x - y)$, where $L \in \mathbb{R}^{d_y \times d_y}$ is learned. For regular grids, interpolation was unnecessary, as dimension-wise factorization with the composite trapezoidal rule allowed the KNO to directly process the training data.

Table 2 demonstrates that the KNO achieves comparable accuracy to the FNO, GNOT, KM, and Transolver across most benchmark problems, with notably superior performance on the challenging Beijing-Air problem, where the KNO is approximately 30% more accurate than FNO and 15% more accurate than the GNOT; the KNO is only 2% more accurate than the Transolver on this problem. The 3D compressible Navier–Stokes problem (Mach 1.0) proves difficult for all methods, highlighting the limitations of current operator learning techniques, with the Transolver achieving the best result by a small 0.5% margin. Interestingly, despite the periodic boundary conditions inherent to this problem, the KNO achieves approximately 16% higher accuracy than FNO, with the best results obtained using the GSM kernel rather than the NS-GSM kernel. The Darcy (triangle) problem also warrants attention. While the GeoFNO appears to slightly outperform KNO on this task[3], the standard errors reported in Table 8 indicate that the results are nearly identical for both methods. Furthermore, as shown in Table 4, the GeoFNO required several million trainable parameters to achieve this level of accuracy, whereas the KNO achieved comparable performance with only approximately 50k parameters. Table 3 further highlights that the best KNO results across most benchmarks were

---

[3]This may be attributed to the simplicity of the mapping from the triangular domain to the unit square used in GeoFNO, which may not be true for other general domains.

Table 3: Kernel and Quadrature Choices. The table lists the integration kernel ("Int."), use of dimension-wise factorization ("Dim. Fac."), and quadrature rule ("Quad."), including trapezoidal ("Trap."), symmetric ("Sym."), and spherical ("Spherical") rules. See Section 2.2.3 for mathematical details and Section A.2.1 for implementation specifics.

| Problem | Int. | Dim. Fac. | Quad. |
|---|---|---|---|
| Burgers' | NS-GSM | Yes | Trap. |
| Beijing-Air | GSM | Yes | Trap. |
| Darcy (PWC) | NS-GSM | Yes | Trap. |
| Darcy (triangle) | NS-GSM | No | Sym. |
| NS-Pipe | NS-GSM | Yes | Trap. |
| NS Mach 1.0 | GSM | Yes | Trap. |
| React.-Diff. | NS-GSM | No | Spherical |

obtained using the NS-GSM kernel, which is anisotropic, non-stationary, and trainable. However, exceptions were observed for the Beijing-Air and NS Mach 1.0 problems, where the GSM kernel outperformed the NS-GSM kernel. This discrepancy may be attributed to difficulties in training the NS-GSM kernel due to the complex loss landscapes associated with these problems, introduced by their inherent noise: Beijing-Air due to it being real-world and NS Mach 1.0 due to its turbulence.

Notably, this underscores another strength of the KNO: in challenging settings, it is straightforward to switch to a simpler kernel, thanks to the KNO's inherent ability to explicitly specify kernels. It is also useful to note that the one-parameter KM performs very well on problems with smooth operator maps, such as the Darcy (triangle) and the reaction-diffusion problem; this method can naturally be viewed as a particular edge case of the KNO, indicating that lower parameter counts and greater architectural simplicity may be important for certain operator learning problems.

## 4.2 Parameter counts

Table 4: Parameter Counts. The table reports the trainable parameter counts for all methods. The entry marked with "**" indicates that the GNOT parameter count was not provided. in Wu et al. (2024).

| Problem | FNO | GNOT | Transolver | KNO |
|---|---|---|---|---|
| Burgers' | 221,889 | 2,843,013 | 382,993 | 43,137 |
| Beijing-Air | 353,217 | 2,182,532 | 2,806,849 | 335,617 |
| Darcy (PWC) | 4,743,937 | 2,183,812 | 2,811,073 | 61,121 |
| Darcy (triangle) | 5,967,619 | 885,062 | – | 49,863 |
| NS-Pipe | 1,188,385 | ** | 2,810,817 | 274,561 |
| NS Mach 1.0 | 14,164,513 | 1,523,587 | 3,791,377 | 31,105 |
| Reaction–Diffusion | 17,746,276 | 886,470 | 372,257 | 32,647 |

The trainable parameter counts for the KNO are presented in Table 4; we did not include the KM which only needed one trainable parameter (but requires storage of the training functions for inference). Except for the Beijing-Air dataset, the KNO consistently required 1-2 orders of magnitude fewer trainable parameters compared to the FNO, GNOT and Transolver while achieving comparable or superior accuracy. This reduction in parameter count did translate to faster training and inference times than the GNOT and Transolver, but the FNO is still faster to train and infer with (see Section 4.3). This parameter count also results in a substantially smaller memory footprint once the model is trained. For instance, with weights stored in fp32, the largest FNO model required approximately 54 MB of storage, the largest GNOT model required approximately 5.8 MB of storage, and the largest Transolver model required 15.2 MB of storage (all for the 3D NS Mach 1.0 problem). In contrast, the KNO required only 0.11 MB of storage for the same problem, highlighting its favorable memory scaling properties compared to the FNO, GNOT, and Transolver. As mentioned previously, we view this as a promising feature for the KNO's use in practical settings where operator surrogates are used. Further, since the KNO has better memory scaling properties, we anticipate that it will prove to be highly efficient in high-dimensional operator learning problems (such as design and inverse problems).

For problems on regular grids and simplicial meshes, integrals can be computed on the fly, eliminating the need for storage of quadrature rules. This makes the KNO particularly appealing from a memory efficiency perspective. We

anticipate that these favorable memory scaling properties will persist for higher-dimensional and larger problems, making KNO an excellent candidate for on-chip surrogate modeling in low-memory environments.

## 4.3 Timings

Table 5: Average Training Time per Epoch (in seconds). Training times were averaged over 100 epochs using mini-batches of size 10 on a NVIDIA GeForce RTX 4080. For KMs we report the time for a single linear system solve instead.

| Problem | FNO | GNOT | Transolver | KM | KNO |
|---|---|---|---|---|---|
| Beijing-Air | 2.56e–3 | 1.00e–2 | 9.60e–3 | 1.49 | 7.46e–3 |
| Darcy (PWC) | 4.44e–3 | 1.49e–2 | 1.53e–2 | 1.49e–2 | 4.56e–3 |
| Reaction-Diffusion | 4.77e–2 | 4.91e–2 | 2.94e–2 | 3.84 | 7.60e–2 |

Table 6: Average Inference Time (in seconds). Inference times were averaged over 100 epochs on a NVIDIA GeForce RTX 4080 with mini-batches of 10.

| Problem | FNO | GNOT | Transolver | KM | KNO |
|---|---|---|---|---|---|
| Beijing-Air | 6.91e–4 | 3.60e–3 | 2.50e–3 | 6.15e–3 | 1.13e–3 |
| Darcy (PWC) | 8.82e–4 | 5.21e–3 | 4.96e–3 | 2.32e–3 | 9.14e–4 |
| Reaction-Diffusion | 1.58e–2 | 2.05e–2 | 1.03e–2 | 6.10e–3 | 2.03e–2 |

We present training times (Table 5) and inference times (Table 6) for the KNO, FNO, GNOT, KM and Transolver across 1D, 2D, and 3D problems. The KNO was implemented in Jax, while the original implementations of FNO and GNOT by their respective authors were in PyTorch. Table 5 shows that the FNO trains nearly twice as fast as KNO, likely due to its use of the FFT and custom CUDA kernels. In contrast, the KNO relies solely on Jax-level optimizations for integral computations, which may contribute to slower training times. The GNOT and Transolver appear comparable to the KNO in training speed, with some evidence suggesting slightly better scaling for the 3D problem.

Table 6 indicates that the FNO is also faster during inference, likely for the same reason. However, the KNO demonstrates faster inference times than the GNOT and Transolver (save for Transolver on the Reaction-Diffusion problem) likely due to its significantly smaller number of trainable parameters. Interestingly, the gap in inference speed between the KNO, GNOT and Transolver does not fully align with the disparity in their parameter counts, potentially pointing to implementation inefficiencies in the KNO. Addressing these inefficiencies may require re-implementing the KNO layers using standard machine learning paradigms, such as convolution layers or attention mechanisms.

## 5 Conclusion

We presented the kernel neural operator (KNO), a simple and transparent architecture that leverages kernel-based deep integral operators discretized by numerical quadrature. By employing highly-expressive, closed-form kernels parametrized by shallow neural networks, the KNO achieved comparable or superior accuracy with far fewer trainable parameters than other neural operators, both on regular and irregular domains. This reduction in parameter count resulted in a significantly smaller memory footprint, making the KNO particularly appealing for surrogate modeling in resource-constrained hardware. Additionally, the KNO demonstrated competitive training and inference times compared to other neural operators, though an analysis of timings relative to parameter counts revealed potential implementation inefficiencies.

For future work, we aim to address these inefficiencies by recasting KNO operations using efficient machine learning paradigms, such as convolution layers and attention mechanisms, which leverage highly optimized CUDA kernels. With the advent of mid-level languages such as Triton Tillet et al. (2019) and high-level languages such as Nvidia's Warp Macklin (2022) that simplify the development of custom GPU kernels, it is feasible to design specialized implementations tailored to the structure of the KNO, enabling further gains through operator-specific fusion and memory-efficient execution. We also plan to explore interpretable lifting and projection operators, problem-specific

architectures tailored to linear operators, and novel quadrature schemes. With respect to quadrature, we foresee utilizing sparse grid or Quasi-Monte carlo quadrature rules to further improve the KNO's efficiency in high dimensional settings. Finally, we save the pursuit of further theoretical results such as approximation rates and sampling estimates for future work. Beyond approximating PDE solution operators, we anticipate that the KNO will be widely applicable to various machine learning tasks, particularly as an on-chip surrogate model in low-memory environments, which will be another focus of our future research.

## Acknowledgements

This work was sponsored by the Sandia National Laboratories Laboratory Directed Research Development (LDRD) program. Sandia National Laboratories is a multi-mission laboratory managed and operated by National Technology & Engineering Solutions of Sandia, LLC (NTESS), a wholly owned subsidiary of Honeywell International Inc., for the U.S. Department of Energy's National Nuclear Security Administration (DOE/NNSA) under contract DE-NA0003525. This written work is authored by an employee of NTESS. The employee, not NTESS, owns the right, title and interest in and to the written work and is responsible for its contents. Any subjective views or opinions that might be expressed in the written work do not necessarily represent the views of the U.S. Government. The publisher acknowledges that the U.S. Government retains a non-exclusive, paid-up, irrevocable, world-wide license to publish or reproduce the published form of this written work or allow others to do so, for U.S. Government purposes. The DOE will provide public access to results of federally sponsored research in accordance with the DOE Public Access Plan.

This work used the Delta system at the National Center for Supercomputing Applications [award OAC 2005572] through allocation [MTH250044] from the Advanced Cyberinfrastructure Coordination Ecosystem: Services & Support (ACCESS) program, which is supported by National Science Foundation grants #2138259, #2138286, #2138307, #2137603, and #2138296.

VS was also supported by Air Force Office of Scientific Research (AFOSR) LRIR grant FA9550-25-1-0042 and National Science Foundation grants SHF 2403379 and DMS 2505986.

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

# A  Appendix

## A.1  Architectural and Computational Details of KNO

### A.1.1  Cross-channel Affine Transformations

Similar to other neural operators Li et al. (2021), each layer of KNO is augmented with a cross-channel affine transformation (commonly referred to as a "pointwise convolution"). This operation, implemented as a dense layer, adds its output to the result of the integral operator. Formally, the integral operators act on and output vectors of function evaluations on $X_Q := \{y_i^q\}_{i \in [N_Q]}$:

$$\tilde{\mathcal{I}}_\ell^p \tilde{\boldsymbol{g}}_\ell = \boldsymbol{W}_\ell \tilde{\boldsymbol{g}}_\ell + (\boldsymbol{b}_\ell)\mathbf{1}_{N_Q}^\top + (\mathcal{I}_\ell^p \tilde{\boldsymbol{g}}_\ell)\big|_{X_Q}, \ \ell \in [L]. \tag{17}$$

Here, $\tilde{\boldsymbol{g}}_\ell \in \mathbb{R}^{p \times N_Q}$ represents evaluations of the function $\boldsymbol{g}_\ell : \Omega \to \mathbb{R}^p$ on $X_Q$, and $\boldsymbol{W}_\ell \in \mathbb{R}^{p \times p}$ and $\boldsymbol{b}_\ell \in \mathbb{R}^p$ are trainable weights. Note that we slightly abuse notation in $(\mathcal{I}_\ell^p \tilde{\boldsymbol{g}}_\ell)\big|_{X_Q}$, as the integral operator acts on the vector $\tilde{\boldsymbol{g}}_\ell$ evaluated at quadrature points rather than a function. The final discretized integral operator outputs values on the training grid $X_T$, which are used to evaluate the loss.

### A.1.2  Lifting and Projection Operators

As with other neural operators, KNO employs standard multilayer perceptrons (MLPs) to parameterize the lifting and projection operators $\mathcal{L}$ and $\mathcal{P}$, which act on discretized inputs. The lifting operator $\mathcal{L}_{X_T} : \mathbb{R}^{N_T} \to \mathbb{R}^p$ maps gridded function values to channel vectors in $\mathbb{R}^p$ using an MLP. Similarly, the projection operator $\mathcal{P}$ combines all $p$ channels of the hidden layers to produce a single approximation of the output function(s). Specifically, the projection operator consists of two consecutive dense layers of width $p$ ($\mathcal{A} : \mathbb{R}^p \to \mathbb{R}^p$) with nonlinear activation functions, followed by a final dense layer of width $d_y$ ($\mathcal{A} : \mathbb{R}^p \to \mathbb{R}^{d_y}$) without an activation function. In all cases, we use the GeLU activation function Hendrycks & Gimpel (2016).

### A.1.3  Details on Quadrature Rules

Table 7: The number of quadrature points we used for each problem.

| Problem | Burgers' | Beijing-Air | Darcy (PWC) | Darcy (triangle) | NS-Pipe | NS Mach 1.0 | React.-Diff. |
|---|---|---|---|---|---|---|---|
| $\mathbf{N_Q}$ | 128 | 168 | 29 | 147 | 129 | 64 | 729 |

For problems on regular grids, we used dimension-wise factorizable kernels with a univariate trapezoidal quadrature rule to perform integration directly on the grid, as reflected in the main results in Table 2. For the Reaction-Diffusion problem, we employed a spherical volume quadrature rule von Winckel (2025), visualized in Figure 5. For the Darcy (triangle) problem, we used the quadrature rule from Freno et al. (2020), described in the main article.

In Table 7, we report the number of quadrature nodes used for each problem. While adaptive, problem-specific quadrature rules could further optimize performance and reduce $X_Q$, we leave such explorations for future work.

Spherical Volume Quadrature Rule

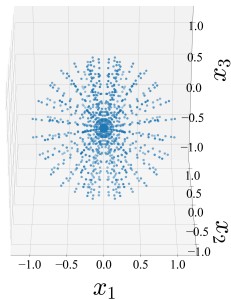

Figure 5: The quadrature rule used for the 3D reaction-diffusion problem.

### A.1.4   Complexity of Computing Integrals via Quadrature

We now discuss the complexity of evaluating the kernel integrals using quadrature and contrast this with the FNO. First, we note that the classic FNO utilizes the FFT and the same training and quadrature grids; thus, given a training grid with $N_T$ grid points, the FFT can be computed in $O(N_T \log N_T)$. In the case of the KNO, we have three distinct scenarios:

1. **Regular grids**: On regular grids, the KNO uses a dimension-wise factorization in conjunction with univariate composite (local) quadrature directly on the training grid. Consequently, the quadrature cost is linear in the number of grid points, i.e., $\Theta(N_T)$ even in high dimensions (high $d_y$). Thus, at least on regular grids, the KNO scales to high dimensions without being afflicted by the curse of dimensionality.

2. **Triangle meshes**: Given $n_q$ quadrature points per triangle and $N_\Omega$ triangles in total, the cost on the triangle mesh is $O(n_q N_\Omega)$. This cost scales exponentially for simplicial meshes in higher dimensions, though this exponential scaling can be combated with sparse grid methods Holtz (2011) and Monte Carlo methods Dick et al. (2013).

3. **Point clouds**: For point clouds, one typically estimates the cost based directly on the total number of quadrature points, since global quadrature is typically used. This cost also scales exponentially with dimension since exponentially more quadrature points are needed to fill hypervolumes in higher dimensions, but can be combated similarly with sparse grids and Monte Carlo (or quasi-Monte Carlo) methods.

### A.1.5   Zero-shot Super-resolution

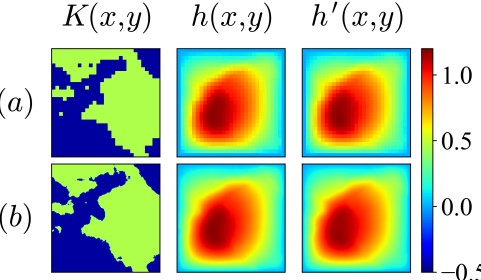

Figure 6: Illustration of zero-shot super-resolution. The KNO was trained on the Darcy (PWC) dataset using a $29 \times 29$ grid (row a) and evaluated at a resolution of $211 \times 211$ (row b). The permeability field input (left), actual pressure field (middle), and predicted pressure field (right) are shown.

The KNO, like the FNO, can achieve zero-shot super-resolution due to its function-space operations at every layer. This capability allows KNO to produce operator solutions at arbitrary resolutions without requiring retraining. Figure 6 demonstrates this property, where KNO was trained on a $29 \times 29$ grid and successfully evaluated on a much finer $211 \times 211$ grid.

## A.2   Hyperparameter Tuning and Training

In this section, we outline the hyperparameter tuning and training protocols for competing models, including FNO, Geo-FNO, and GNOT, to ensure a fair and consistent comparison with KNO.

### A.2.1   KNO

For the KNO, all trainable parameters associated with kernels were initialized by sampling from $\mathcal{N}(1, 0.01)$, followed by a softplus transform to ensure that all kernel shape parameters remained positive. The number of Gaussians in the mixture for all GSM/NS-GSM kernels was fixed at $Q = 2$. For nonstationary kernels, the shallow SELU networks used had a width of 8.

For hyperparameters that were tuned, we performed a grid search using a random seed different from the one used for collecting the final results. The hyperparameters included:

- **Number of integration layers:** searched over $\{2, 3, 4\}$.

- **Number of channels:** searched over $\{8, 16, 32, 64, 128, 256\}$.

- **Type of integration kernel:** details provided in Appendix A.7.1.

- **Type of interpolation kernel:** searched between an anisotropic Gaussian kernel and an isotropic Gaussian kernel, with better results observed using the anisotropic kernel.

- **Number of quadrature nodes:** tuned based on the specific problem, with details provided in Appendix A.1.3.

### A.2.2 Alternative Operator Models

We performed a grid search to identify the best-performing hyperparameters for each alternate model, and the details are provided below.

**FNO**[4]: The hyperparameters included the number of modes, varied over $\{8, 10, 12, 16, 20\}$, the number of channels for channel lifting, varied over $\{8, 16, 32, 64, 128, 256\}$, and the number of Fourier layers, varied over $\{2, 3, 4\}$. We used GELU activation, which is the default choice in the official library.

**Geo-FNO**[5]: For this model, we searched over the one-dimensional resolution of the uniform tensor-product grid that is mapped to (denoted $s$ in their codebase), with the maximum number of modes set for each case. We then searched for the optimal number of modes. For the Darcy Triangular problem, $s$ was varied over $\{15, 20, 25, 30, 35\}$ and the modes over $\{8, 10, 12, 16\}$. For the Diffusion-Reaction problem, $s$ was varied over $\{8, 10, 12, 14\}$ and the modes over $\{4, 6, 8\}$.

**GNOT**[6]: The hyperparameters included the number of attention layers, varied over $\{2, 3, 4, 5\}$, the dimensions of the embeddings, varied over $\{8, 16, 32, 64, 128, 256\}$, and the inclusion of mixture-of-expert-based gating, specified as either $\{yes, no\}$. We used GELU activation, which is the default choice in the official library.

**Transolver**[7]: The hyperparameters included the number of attention layers, varied over $\{4, 6, 8\}$, the dimensions of the embeddings, varied over $\{32, 64, 128\}$, the number of heads, varied over $\{4, 8\}$, and the number of slices, varied over $\{32, 64\}$. We used GELU activation, which is the default choice in the official library.

**KM**: We report the best result on a given dataset from the choice of Matern Kernels with degree of freedom $\nu = \{1/2, 3/2, 5/2, 13/2\}$ and a Gaussian kernel. In each case the scale parameter was manually tuned.

For the NS-Pipe Flow dataset, we used the result from Wu et al. (2024) for GNOT, which was collected under the same hyperparameter search we employed and Transolver, which reports itself. For FNO, we cite the authors' result from Li et al. (2023). To ensure fairness, all models (including KNO) for this dataset were trained for 500 epochs. For all other datasets, FNO and GNOT were trained for $10,000$ epochs with batch sizes of $100$ to ensure convergence. Initial learning rates were selected from $\{10^{-5}, 5 \times 10^{-5}, 10^{-4}, 4 \times 10^{-3}, 10^{-3}\}$. All models were trained on the NCSA Delta GPU cluster[8] using NVIDIA A100 and A40 GPUs.

### A.3 Descriptions of Benchmark Problems Defined on Regular Domains

In this section, we describe problems on regular domains in greater detail (where not sufficiently described in the main article).

### A.3.1 1D Burgers' Equation

We first considered Burgers' equation in one dimension with periodic boundary conditions:

$$\frac{\partial u}{\partial t} + u \frac{\partial u}{\partial x} = \nu \frac{\partial^2 u}{\partial x^2}, \quad x \in (0, 1), \quad t \in (0, 1),$$

with the viscosity coefficient fixed to $\nu = 0.1$. Specifically, we learned the mapping from the initial condition $u(x, 0) = u_0(x)$ to the solution $u(x, t)$ at $t = 1$, *i.e.*, $\mathcal{G} : u_0 \mapsto u(\cdot, 1)$. The input functions $u_0$ were generated by sampling $u_0 \sim \mu$, where $\mu = \mathcal{N}(0, 625(-\Delta + 25I)^{-2})$ with periodic boundary conditions, and the Laplacian $\Delta$ was numerically approximated on $X_T$. The solution was generated as described in (Li et al., 2021, Appendix A.3.1). The full spatial resolution of this dataset was 8192, but the models were trained and evaluated on input-output function

---

[4]https://github.com/neuraloperator/neuraloperator
[5]https://github.com/neuraloperator/Geo-FNO
[6]https://github.com/HaoZhongkai/GNOT
[7]https://github.com/thuml/Transolver
[8]https://www.ncsa.illinois.edu/research/project-highlights/delta/

pairs both defined on the same downsampled 128 grid (as were the errors). 1000 examples were used for training and 200 for testing.

### A.3.2  1D Beijing Air problem

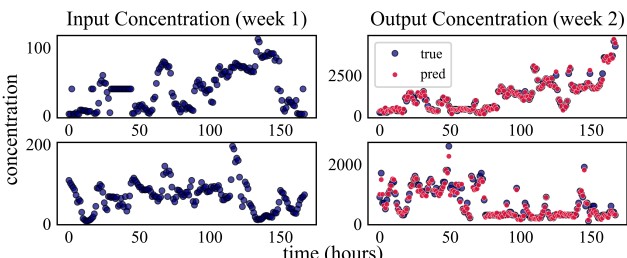

Figure 7: An input pollutant concentration (left) and the corresponding ground truth and KNO predictions of output pollutant concentration (right) for the Beijing-Air problem.

This problem was fully described in the main article. However, here we also present Figure 7, which shows the KNO predictions as compared to the ground truth on the Beijing-Air dataset. Despite the noise in the dataset, the KNO gives the best performing result of the neural operators tested.

### A.3.3  2D Darcy Flow (PWC)

We used KNOs to learn an operator $\mathcal{G} : K \mapsto h$ associated with 2D Darcy flow

$$-\nabla \cdot (K(x,y)\nabla h(x,y)) = f(x,y), \quad (x,y) \in \Omega.$$

on the $\Omega = [0,1]^2$. The permeability field was generated via $K = \psi(\mu)$, where $\mu \sim \mathcal{N}(0, (-\Delta + 9I)^{-2})$, and $\psi$ is a function that pointwise converts all non-negative values to 12 and all negative values to 3. Accordingly we referred to this problem as "Darcy (PWC)". Both problems used 1000 training functions and 200 test functions. The Darcy (PWC) training functions were computed on a $421^2$ grid and subsampled to a $29^2$ grid Lu et al. (2022).

### A.3.4  3D Compressible Navier–Stokes (NS) Equations.

The KNO was also tested on a problem involving the 3D compressible NS equations:

$$\partial_t \rho + \nabla \cdot (\rho \mathbf{v}) = 0,$$
$$\rho (\partial_t \mathbf{v} + \mathbf{v} \cdot \nabla \mathbf{v}) = -\nabla p + \eta \Delta \mathbf{v} + (\zeta + \eta/3)\nabla(\nabla \cdot \mathbf{v}),$$
$$\partial_t \left[ \epsilon + \frac{\rho v^2}{2} \right] + \nabla \cdot \left[ \left( \epsilon + p + \frac{\rho v^2}{2} \right) \mathbf{v} - \mathbf{v} \cdot \sigma' \right] = 0 \tag{18}$$

with periodic boundary conditions on the unit hypercube $[0,1]^3$. Here $\rho$ is the mass density, $\mathbf{v}$ is the velocity, $p$ is the gas pressure, $\epsilon = p/(\Gamma - 1)$ is the internal energy, $\Gamma = 5/3$, $\sigma'$ is the viscous stress tensor, and $\eta, \zeta$ are the shear and bulk viscosity, respectively. The behavior of the fluid is affected by the Mach number $M = |v|/c_s$, where $c_s = \sqrt{\Gamma p/\rho}$. We considered the high Mach number case ($M = 1.0$), where the fluid behavior is complex and highly compressible. The input and output functions were discretized on a uniform grid of size $64 \times 64 \times 64$. This data was made available through PDEBench Takamoto et al. (2022), which offers many widely used benchmarks for scientific machine learning.

### A.4  Descriptions of Benchmark Problems Defined on Irregular Domains

In this section, we describe problems on irregular domains in greater detail (where not sufficiently described in the main article).

### A.4.1  Darcy (triangular)

We also examined a Darcy flow problem where the input and output functions were discretized on an irregular spatial domain. Specifically, as in Lu et al. (2022), we learned the mapping from the Dirichlet boundary condition to the pressure field over the entire domain, *i.e.*, the operator $\mathcal{G} : h(x,y)|_{\partial\Omega} \mapsto h(x,y)$. Here $K(x,y) = 0.1$ and $f = -1$. The input functions $h(x,y)|_{\partial\Omega}$ were generated as follows. First, we generated $\tilde{h}(x) \sim \mathcal{GP}(0, \mathcal{K}(x,x'))$, $\mathcal{K}(x,x') =$

$\exp[-\frac{(x-x')^2}{2l^2}]$, where $l = 0.2$ and $x, x' \in [0, 1]$. We then simply evaluated $\tilde{h}(x)$ at the $x$-coordinates of the boundary points of each unstructured mesh to obtain $h(x, y)\big|_{\partial\Omega}$. The Matlab PDE Toolbox was used both to generate unstructured meshes and numerical solutions Lu et al. (2022). This problem utilized an 861 vertex unstructured mesh with 120 points lying on the boundary; see Lu et al. (2022) with 1900 training examples and 100 test examples.

### A.4.2 3D Reaction-Variable-Coefficient-Diffusion

Finally, we investigated a 3D reaction-diffusion problem in the unit ball, (*i.e.*, the interior of the unit sphere) where a chemical with concentration $c(y, t)$ is governed by:

$$\frac{\partial c}{\partial t} = k_{\text{on}}\,(R - c)\,c_{\text{amb}} - k_{\text{off}}\,c + \nabla \cdot (K(y)\nabla c)\,, \ y \in \Omega, \ t \in [0, 0.5],$$

where $y = (y_1, y_2, y_3)$ and $K(y)\frac{\partial c}{\partial n} = 0$ on $\partial\Omega$. Here, $R = 2.0$ throttles the reaction, and the $k_{\text{on}}$ and $k_{\text{off}}$ are discontinuous reaction constants that introduce a sharp solution gradient at $y_1 = 1.0$:

$$k_{\text{on}} = \begin{cases} 2, & y_1 \le 1.0, \\ 0, & \text{otherwise,} \end{cases} \qquad k_{\text{off}} = \begin{cases} 0.2, & y_1 \le 1.0, \\ 0, & \text{otherwise.} \end{cases}$$

The diffusion coefficient is also a spatially varying function with a steep gradient given by:

$$K(y) = B + \frac{C}{\tanh(A)}\left((A - 3)\tanh(8x - 5) - (A - 15)\tanh(8x + 5) + A\tanh(A)\right),$$

where $A = 9$, $B = 0.0215$, and $C = 0.005$. $c_{\text{amb}} = (1 + \cos(2\pi y_1)\cos(2\pi y_2)\sin(2\pi y_3))e^{(-\pi t)}$ is a background source of chemical accessible for reaction. We set the initial condition to be $c(y, 0) \sim \mathcal{U}(0, 1)$, and learned the solution operator $\mathcal{G} : c(y, 0) \to c(y, 0.5)$. The PDE was solved on 4325 collocation points using a 4th-order accurate RBF-FD solver Shankar & Fogelson (2018) to generate 1000/200 train and test input/output function pairs, respectively.

### A.5 Results including standard errors

Table 8: Percent $\ell_2$ Relative Errors and Standard Errors.

| Problem | FNO | GNOT | Transolver | KM | KNO |
|---|---|---|---|---|---|
| Burgers $\nu = 0.1$ | **0.276 ± 0.004** | 0.89 ± 0.014 | 1.077 ± 0.101 | 2.831 | 0.574 ± 0.011 |
| Beijing-Air | 55.33 ± 0.18 | 40.3 ± 0.21 | 41.68 ± 0.332 | 50.982 | **24.941 ± 0.161** |
| Darcy (PWC) | 1.79 ± 0.025 | 2.58 ± 0.034 | 1.989 ± 0.0278 | 3.064 | **1.512 ± 0.012** |
| Darcy (triangle) | **0.043 ± 0.003** | 0.111 ± 0.013 | – | 0.033 | 0.045 ± 0.002 |
| NS-Pipe | 0.67 | 0.47 ± 0.02 | 0.33 ± 0.02 | 2.742 | **0.317 ± 0.0150** |
| NS Mach 1.0 | 58.05 ± 4.77 | 81.5 ± 2.18 | **48.127 ± 0.483** | 54.150 | 48.636 ± 0.073 |
| React.-Diff. | 6.68e-3 ± 2.77e-4 | 4.47e-3 ± 9.45e-5 | 8.10e-3 ± 1.89e-5 | **8.75e-05** | 9.20e-4 ± 1.02e-4 |

Table 8 replicates Table 2, but also shows standard errors for each result. As mentioned previously, the Darcy (triangle) result is notable in that the (Geo)FNO and KNO results match almost exactly.

### A.6 Dataset Summary

Table 9: Training Dataset Summary.

| Geometry | Benchmarks | #Dim | #Mesh | Input | Output | #Dataset |
|---|---|---|---|---|---|---|
| Regular Grid | Beijing-Air | 1D | 168 | SO2, CO, PM2.5, PM10 | CO conc. | (1000, 200) |
| | Burgers | 1D | 168 | Init. velocity | Final velocity | (5000, 1000) |
| | Darcy | 2D | 841 | Permeability | Pressure | (1000, 200) |
| | Navier-Stokes Mach 1.0 | 3D | 262,144 | Init. velocity | Final velocity | (90,10) |
| Structured Mesh | Pipe | 2D | 16,641 | Structure | Final velocity | (1000, 200) |
| | Darcy (triangle) | 2D | 120/861 | Boundary Condition | Pressure | (1900, 100) |
| Point Cloud | Reaction-Diffusion | 3D | 4,325 | Init. conc. | Final conc. | (1000, 200) |

Table 9 summarizes important features of the datasets used in this work. "#Mesh" indicates the number of sample locations. For the Darcy (triangle) problem, we map from 120 boundary vertices to 861 interior and boundary vertices in total. The "#Dataset" column indicates the number of training and test functions.

## A.7 Ablation Studies

In this section, we present experiments that analyze the sensitivity of the KNO to various hyperparameter choices. All results are reported as the percent $\ell_2$ relative error on generalization, averaged over four random model initializations.

### A.7.1 Impact of Kernel Properties on Performance

Table 10: Percent $\ell_2$ relative errors (averaged over four random seeds) for different kernels, with the best results highlighted in bold.

| **Problem** | Gaussian | Gibbs | GSM | NS-GSM |
|---|---|---|---|---|
| Burgers | 1.061 | 1.169 | 0.783 | **0.574** |
| Beijing-Air | 27.748 | 31.477 | **24.941** | 26.540 |
| Darcy (PWC) | 2.085 | 2.385 | 1.792 | **1.549** |
| Darcy (triangle) | 0.264 | 0.271 | 0.102 | **0.045** |
| NS-Pipe | 0.749 | 0.697 | 0.719 | **0.588** |

In this section, we describe an ablation study over integration kernels with the goal of isolating the effect of non-stationarity and evaluating the importance of the number of trainable parameters. We compare the one-parameter Gaussian kernel, the Gibbs kernel (i.e., nonstationary Gaussian kernels), Gaussian Spectral Mixture (GSM) kernel, and the Nonstationary-Gaussian Spectral Mixture (NS-GSM) kernel, as shown in Table 10. The trainable parameter counts for the Gaussian, Gibbs, GSM, and NS-GSM kernels used in this 1D example were 1, 25, 6, and 70, respectively, showcasing a spectrum of increasing expressivity (at least in theory).

Surprisingly, the KNO with just the one-parameter Gaussian kernel delivers strong performance, outperforming both GNOT and FNO on the challenging Beijing-Air problem ($\sim 28\%$ relative error compared to $\sim 40\%$ for GNOT and $\sim 55\%$ for FNO). It also surpassed GNOT and Transolver on the Darcy (PWC) problem (1.79% relative error compared to 2.58%) and matched the FNO's result with the GSM kernel. Additionally, the KNO with the GSM kernel demonstrated superior performance over the GNOT and Transolver on the Burgers problem (0.783 vs. 0.89) and 1.077) respectively, and the GNOT on the Darcy (triangle) problem (0.10 vs. 0.11), while also matching FNO's result on the Darcy problem. Clearly, Table 10 shows that increasing kernel expressivity leads to improved results, but primarily as the problem itself becomes more challenging.

Table 11: Percent $\ell_2$ relative errors (averaged over four random seeds) for different kernels, with the best results highlighted in bold.

| **Problem** | Gaussian | Matern $C_2$ | Matern $C_6$ | Wendland $C_2$ | Wendland $C_6$ |
|---|---|---|---|---|---|
| Burgers | 1.061 | 1.427 | 1.489 | 1.197 | 1.168 |
| Beijing-Air | 27.748 | 35.726 | 37.0170 | 24.167 | 24.170 |
| Darcy (PWC) | 2.085 | 3.560 | 3.666 | 2.138 | 2.143 |

In addition to testing kernels with global support and infinite smoothness, we also explored the use of compactly-supported kernels and kernels of finite smoothness. The Wendland Kernel Wendland (1995) possesses both these features, while the Matern kernel only has the latter. The results are presented in table 11. In all cases, the compactly-supported Wendland Kernels outperformed the Matern kernels and performed on-par with Gaussian kernels. In the Beijing-Air problem, the Wendland kernel outperforms the Gaussian kernel; in fact it outperforms all other models, suggesting sparsity is a desirable trait, which can be leveraged to achieve speedups via sparse-matrix operations (when these are supported by the underlying libraries). Otherwise, the Gaussian kernel tends to perform better overall as reflected by its performance on the Burgers and Darcy (PWC) problem.

### A.7.2 Impact of Matrix-valued Kernel Structure on Performance

In this section, we experiment with a symmetric banded matrix-valued kernel on the Darcy (PWC) problem and the Burgers' problem, incorporating a hyperparameter $k$ that reflects an additional number of non-zero off diagonal entries incorporated into the MVK (mirrored across the matrix), each with a unique set of kernel parameters. Thus if $k$ is equal to the channel dimension $p$, the MVK is fully dense. These KNOs tended to be slightly more expressive with potential for further improvement given more careful tuning of the optimization procedure. From Table 12, we can

glean that a denser MVK is more helpful with respect to errors when the channel dimension is small than it is when the channel dimension is large.

We also trained models with the Linear Model of Coregionalization (equation 20, section 4.2.1) Álvarez et al. (2012) and the Multi-output Spectral-Mixture kernel Parra & Tobar (2017), but the results were not competitive and so we omit them here.

Table 12: Relative errors for different kernels across channel dimensions and $k$ values.

| Problem | Channels | $k=0$ | $k=1$ | $k=2$ | $k=4$ | $k=8$ | $k=C$ |
|---|---|---|---|---|---|---|---|
| Burgers' | 16 | 1.247 | 1.101 | 1.068 | 0.939 | **0.816** | 0.823 |
| | 32 | 0.718 | 0.654 | 0.595 | 0.550 | **0.504** | 0.509 |
| | 64 | 0.574 | 0.546 | 0.555 | 0.563 | **0.527** | 2.161 |
| Darcy (PWC) | 16 | 1.595 | 1.573 | 1.496 | 1.466 | **1.462** | 1.611 |
| | 32 | 1.483 | **1.445** | 1.455 | 1.464 | 1.625 | 4.572 |
| | 64 | 1.689 | 1.616 | **1.614** | 1.632 | 1.649 | 1.724 |

### A.7.3   Impact of Model Size on Performance

Table 13: Ablation study of the KNO on the number of integration layers (depth) and channel dimension for the Darcy (PWC) problem. The base model uses 4 integration layers, 64 channels, and the NS-GSM kernel. The best results are highlighted in bold.

(a) Depth $L$: the number of layers.

| Depth | 2 | 3 | 4 | 5 | 6 |
|---|---|---|---|---|---|
| % rel $\ell_2$ | 2.380 | 1.825 | 1.549 | **1.534** | 1.569 |

(b) The number of latent channels $C$.

| Channels | 8 | 16 | 32 | 64 | 128 | 256 |
|---|---|---|---|---|---|---|
| % rel $\ell_2$ | 3.072 | 2.230 | 1.643 | **1.549** | 1.868 | 2.282 |

This section examines the effect of channel dimension and the number of integration layers on KNO's performance, as shown in Tables 13 and 14. The goal of this ablation study was to understand how increasing model size influenced scalability and accuracy, and to identify optimal configurations for different datasets. Experiments were conducted on the Darcy (PWC) dataset and the Beijing-Air dataset to analyze these effects.

Table 14: Ablation study of KNO on the number of integration layers (depth) and channel dimension for the Beijing-Air problem. The base model uses 4 integration layers, 256 channels, and the GSM kernel. The best results are highlighted in bold.

(a) Depth $L$: the number of layers.

| Depth | 2 | 3 | 4 | 5 | 6 |
|---|---|---|---|---|---|
| % rel $\ell_2$ | 28.93 | 24.94 | 22.15 | 18.37 | **15.52** |

(b) The number of latent channels $C$.

| Channels | 8 | 16 | 32 | 64 | 128 | 256 | 512 |
|---|---|---|---|---|---|---|---|
| % rel $\ell_2$ | 58.23 | 54.93 | 49.59 | 41.89 | 32.16 | 22.15 | **13.67** |

In general, we observed diminishing returns in performance improvements beyond a certain model size. For the Darcy (PWC) dataset, increasing the channel dimension beyond 64 and the number of integration layers beyond 5 did not improve accuracy. In contrast, for the Beijing-Air dataset, scaling the model size consistently improved performance, with the best results achieved at a channel dimension of 512 and 6 integration layers. This difference likely reflects the higher complexity and variability of the Beijing-Air dataset compared to Darcy (PWC).

Table 15: Ablation study of KNO on the number of integration layers (depth) and channel dimension for the NS-Pipe problem. The base model uses 5 integration layers, 64 channels, and the NS-GSM kernel. The best results are highlighted in bold.

(a) Depth $L$: the number of layers.

| Depth | 4 | 5 | 6 | 7 | 8 |
|---|---|---|---|---|---|
| % rel $\ell_2$ | 0.434 | 0.386 | 0.401 | 0.352 | **0.348** |

(b) The number of latent channels $C$.

| Channels | 16 | 32 | 64 | 128 | 192 | 256 |
|---|---|---|---|---|---|---|
| % rel $\ell_2$ | 1.031 | 0.579 | 0.386 | 0.360 | 0.350 | **0.343** |

These findings suggest that while larger models may be beneficial for complex datasets, careful tuning of model size is necessary to avoid over-parameterization and diminishing returns, particularly for simpler datasets.

### A.7.4 Impact of Integration Layers and Pointwise Convolution

Table 16: KNO ablation study on removing the pointwise convolution or integration layer, with a fixed architecture otherwise. Each number reported represents the percent $\ell_2$ relative error.

| Problem | No Pointwise Conv | No integral | Full Model |
|---|---|---|---|
| Darcy (PWC) | 1.915 | 17.058 | 1.512 |
| NS-Pipe | 0.792 | 14.870 | 0.588 |

This section examines the role of integration layers and pointwise convolution in the KNO's performance, as shown in Table 16. The goal of this ablation study is to evaluate the individual contributions of these components to KNO's accuracy and identify their relative importance for different datasets. Experiments were conducted on the Darcy (PWC) and NS-Pipe datasets to analyze how removing these components affects accuracy.

The results clearly demonstrate the necessity of integration kernels, as removing the integration layer reduces model accuracy by an order of magnitude. For example, on the Darcy (PWC) dataset, removing the integration layer increased the relative error from $1.512\%$ to $17.058\%$, while removing the pointwise convolution resulted in a *much* smaller increase to $1.915\%$. Similar trends were observed for the NS-Pipe dataset, where removing the integration layer again caused over a tenfold increase in error. These findings suggest that integration layers are critical for capturing complex operator mappings, while pointwise convolution provides only marginal additional benefits.

### A.7.5 Impact of Quadrature Type and Resolution

Table 17: Experiments analyzing the impact of the number of quadrature nodes ($\mathbf{N_Q}$) on accuracy. The best results are highlighted in bold.

(a) Effect of increasing resolution of the Gauss-Legendre quadrature rule on the 1D Burgers' problem. The base model uses a NS-GSM kernel with 4 integration layers and a channel dimension of 64.

| $\mathbf{N_Q}$ | 8 | 16 | 32 | 64 | 96 | 128 | 196 | 256 |
|---|---|---|---|---|---|---|---|---|
| % rel $\ell_2$ | 26.20 | 25.00 | 6.88 | 1.25 | 0.651 | **0.540** | 0.671 | 0.635 |

(b) Effect of increasing quadrature resolution on the 2D Darcy (triangle) problem. The base model uses a GSM kernel with 4 integration layers and a channel dimension of 64.

| $\mathbf{N_Q}$ | 3 | 12 | 27 | 48 | 75 | 108 | 149 | 192 |
|---|---|---|---|---|---|---|---|---|
| % rel $\ell_2$ | 34.82 | 7.596 | 1.528 | 0.551 | 0.239 | 0.131 | **0.102** | 0.111 |

This section examines the relationship between quadrature type, resolution, and accuracy, as shown in Table 17. The goal of this ablation study is to evaluate how different quadrature rules and resolutions affect the KNO's accuracy and computational efficiency, and to identify optimal configurations for specific datasets. Experiments were conducted on the Darcy (triangle) and Burgers' datasets to analyze these effects. For the Darcy (triangle) problem, we used a quadrature rule from Freno et al. (2020) (described in the main article), while for the Burgers' problem, we employed a Gauss-Legendre rule. For the latter, it is worth noting that our main results in Table 2 used a trapezoidal quadrature rule, which allowed us to omit interpolation and perform integration directly on the grid where the data lay. In this study, however, we used a GSM interpolant to project the functions onto the Gauss-Legendre nodes. For the same number of quadrature nodes, the Gauss-Legendre rule slightly outperformed the trapezoidal rule, achieving $0.540\%$ relative error compared to $0.588\%$. The Darcy (triangle) problem also shows improvements when increasing the accuracy of the quadrature rule. However, interestingly, both sets of results show that while increasing the number of quadrature points initially improves accuracy, using too many points eventually leads to a degradation in performance. These findings suggest that both the quadrature type and the number of quadrature points should be carefully tailored to the specific requirements of each problem such as smoothness, dimension, and geometry. As mentioned in the main article, using lower-order quadrature rules can introduce numerical damping, which can be beneficial or harmful depending on the smoothness of the integrands.

### A.7.6   Impact of Dimension-wise Factorizations

Table 18: Comparison of training time (using mini-batches of size 10) and performance between dimension-wise factorized kernels and non-factorized kernels.

| **Problem** | Dimension-wise | | Full | |
|---|---|---|---|---|
| | Time | % rel $\ell_2$ | Time | % rel $\ell_2$ |
| Darcy (PWC) | 4.56e-3 | 1.549 | 4.88e-2 | 1.535 |

This section examines the effect of dimension-wise factorization on the KNO's accuracy and training speed, as shown in Table 18. The goal of this ablation study is to evaluate the trade-offs between computational efficiency and accuracy when using dimension-wise factorization compared to full $n$-dimensional quadrature. Dimension-wise factorization allows the use of univariate quadrature rules, whereas models without factorization require $n$-dimensional quadrature rules. Experiments were conducted on the 2D Darcy (PWC) problem to analyze these differences.

The results indicate that while the model with dimension-wise factorization experiences only a slight degradation in accuracy, it is an order of magnitude faster to train compared to the non-factorized model. For example, the factorized model achieved a relative error of $1.549\%$, compared to $1.535\%$ for the non-factorized (full) model, while reducing training time from $4.88 \times 10^{-2}$ seconds per epoch to $4.56 \times 10^{-3}$ seconds per epoch. These findings highlight the trade-off between computational efficiency and accuracy, suggesting that dimension-wise factorization is particularly advantageous for problems where training speed is a priority.

Interestingly, while the 2D Darcy (PWC) problem showed only a slight accuracy degradation with dimension-wise factorization, similar experiments on higher-dimensional problems may reveal stronger dependencies on factorization. Future work could explore hybrid approaches that balance factorization with accuracy preservation for more complex problems.

## B   Universal Approximation

### B.1   Infinite-Dimensional Case

The following is a restatement and proof of Theorem 3.1.

**Theorem 3.1.** *Let $\Omega \subset \mathbb{R}^d$ be compact, and let $A \subset (L^2(\Omega; \mathbb{R}), \|\cdot\|_{L^2(\Omega)})$ be compact. Let $\mathcal{G} : A \to (L^2(\Omega; \mathbb{R}), \|\cdot\|_{L^2(\Omega)})$ be a continuous operator. For any $\epsilon > 0$, there exists a KNO $\mathcal{H} : A \to L^2(\Omega; \mathbb{R})$ of the form* (2) *with continuous positive-definite kernels $K_{i_\ell,j_\ell}^{(\ell)}$ such that*

$$\sup_{f \in A} \|\mathcal{H}[f] - \mathcal{G}[f]\|_{L^2(\Omega)} < \epsilon. \tag{19}$$

*Proof.* The structure of this proof is based on the approach of Kovachki et al. (2021). However, we exclusively use diagonal kernels and do not assume that $b_l$ depends on $x$. We use the $L^2$ inner product, and all $L^p$ norms are over $\Omega$ unless otherwise noted.

Let $\epsilon > 0$ be arbitrary. Let $K(x, y)$ be an arbitrary continuous positive-definite kernel on $\Omega$, i.e.

$$\int_\Omega \int_\Omega f(x) K(x, y) g(y) \, dx \, dy \geq 0 \qquad \text{for any } f, g \in L^2(\Omega). \tag{20}$$

By Mercer's theorem, there exist $\{\lambda_k\}_{k \in \mathbb{N}} \subset \mathbb{R}_+$ and $\{\psi_k\}_{k \in \mathbb{N}} \subset L^2(\Omega)$ such that

$$\int_\Omega K(x, y) \psi_k(y) \, dy = \lambda_k \psi_k(x), \qquad K(x, y) = \sum_{k \in \mathbb{N}} \lambda_k \psi_k(x) \psi_k(y),$$

and $\{\psi_k\}_{k \in \mathbb{N}}$ forms an orthonormal basis of $L^2(\Omega)$ with respect to $\| \cdot \|_{L^2}$. Now let

$$\mathcal{G}_N = \mathcal{T}_N \circ \mathcal{G} \circ \mathcal{T}_N$$

where for any $f \in A$,

$$\mathcal{T}_N[f] = \sum_{n \in [N]} \langle f, \psi_n \rangle \, \psi_n \tag{21}$$

i.e. $\mathcal{T}_N$ is the projection onto the first $N$ Mercer eigenfunctions. Note that for any $N \in \mathbb{N}$, $f \in A$, and $\mathcal{H} : A \to L^2(\Omega; \mathbb{R})$ we have

$$\|\mathcal{H}[f] - \mathcal{G}[f]\|_{L^2} \leq \|\mathcal{H}[f] - \mathcal{G}_N[f]\|_{L^2} + \|\mathcal{G}_N[f] - \mathcal{G}[f]\|_{L^2} \, .$$

Since $\mathcal{G}$ is continuous, $\mathcal{T}_N$ is a projector, and $A$ is compact, then there exists $N \in \mathbb{N}$ such that

$$\sup_{f \in A} \|\mathcal{G}_N[f] - \mathcal{G}[f]\|_{L^2} < \epsilon/2,$$

and all that remains is to construct $\mathcal{H}$ such that

$$\sup_{f \in A} \|\mathcal{H}[f] - \mathcal{G}_N[f]\|_{L^2} < \epsilon/2 \, . \tag{22}$$

We define

$$\mathcal{B}_N : \text{span}\{\psi_n\}_{n \in [N]} \to \mathbb{R}^N, \qquad (\mathcal{B}_N[f])_n \equiv \langle f, \psi_n \rangle, \qquad \mathcal{B}_N^{-1}[c](x) = \sum_{n \in [N]} c_n \psi_n(x) \tag{23}$$

along with $\widehat{\mathcal{G}}_N : \mathbb{R}^N \to \mathbb{R}^N$ as

$$\widehat{\mathcal{G}}_N = \mathcal{B}_N \circ \mathcal{G}_N \circ \mathcal{B}_N^{-1},$$

which gives

$$(\mathcal{B}_N^{-1} \circ \widehat{\mathcal{G}}_N \circ \mathcal{B}_N)[f] = \mathcal{G}_N[f] \tag{24}$$

for each $f \in A$. All that remains is to approximate $\mathcal{B}_N$ and $\mathcal{B}_N^{-1}$ with nonlinear operators and to approximate $\widehat{\mathcal{G}}_N$ as a sequence of integral operators

$$(\mathcal{I}_q^p)_L \circ \sigma \circ \cdots \circ \sigma \circ (\mathcal{I}_q^p)_0 \, . \tag{25}$$

as in (2)-(3). We will address $\widehat{\mathcal{G}}_N$ here and consider $\mathcal{B}_N$ and $\mathcal{B}_N^{-1}$ in the following two lemmas.

**Approximation of $\widehat{\mathcal{G}}_N$:** $\widehat{\mathcal{G}}_N$ is a continuous finite-dimensional map, and since $A$ is compact, then the domain $\mathcal{B}_N(\mathcal{T}_N A) = D \subset \mathbb{R}^N$ of $\widehat{\mathcal{G}}_N$ is also compact, where the operators apply elementwise on sets. Therefore, by universal approximation theorem for multilayer perceptrons (MLPs), for any $\tilde{\epsilon} > 0$ there exists an MLP $\overline{\mathcal{G}}$ such that

$$\sup_{x \in D} \|\overline{\mathcal{G}}(x) - \widehat{\mathcal{G}}_N(x)\|_{\ell^\infty(\mathbb{R}^N)} < \tilde{\epsilon} \, . \tag{26}$$

We can write

$$\overline{\mathcal{G}}_N(\boldsymbol{x}) = \boldsymbol{W}^{(1)}\sigma(\boldsymbol{W}^{(0)}\boldsymbol{x} + \boldsymbol{b}^{(0)}) \tag{27}$$

for some $p \in \mathbb{N}$, $W^{(0)} \in \mathbb{R}^{p \times N}$, $W^{(1)} \in \mathbb{R}^{N \times p}$, and $b^{(0)} \in \mathbb{R}^p$. Since $K(x,y) \equiv 0$ satisfies (20), then (27) defines a sequence of integral operators (25) with $L = 1$, $p = N$, and $\boldsymbol{K}^{(0)} = \boldsymbol{K}^{(1)} = \boldsymbol{0}_{N \times N}$ where we interpret the inputs and outputs of $(\widetilde{\mathcal{I}}_q^p)_0$ and $(\widetilde{\mathcal{I}}_q^p)_1$ as constant functions $g^{(0)} : \Omega \to D$, $g^{(1)} : \Omega \to D'$. This interpretation is justified in light of Lemmas B.1 and B.2.

**Construction of $\mathcal{H}$:** Select $\mathcal{P}$, $\widehat{\mathcal{G}}_N$, and $\mathcal{L}$ as follows.

1. Choose $\mathcal{P}$ from Lemma B.2 corresponding to an approximation accuracy of $\epsilon/4$. Since $\mathcal{P}$ is continuous, then there exists $\delta_\epsilon^{(1)}$ such that $\|\mathcal{P}[\boldsymbol{x}] - \mathcal{P}[\boldsymbol{y}]\|_{L^2} \leq \epsilon/8$ whenever $\|\boldsymbol{x} - \boldsymbol{y}\|_{\ell^\infty} \leq \delta_\epsilon^{(1)}$.

2. Choose $\overline{\mathcal{G}}_N$ corresponding to an approximation accuracy of $\tilde{\epsilon} = \delta_\epsilon^{(1)}$. Since $\overline{\mathcal{G}}_N$ is continuous, there exists $\delta_\epsilon^{(2)}$ such that $\|\overline{\mathcal{G}}_N[\boldsymbol{x}] - \overline{\mathcal{G}}_N[\boldsymbol{y}]\|_{\ell^\infty} \leq \delta_\epsilon^{(1)}$ whenever $\|\boldsymbol{x} - \boldsymbol{y}\|_{\ell^\infty} \leq \delta_\epsilon^{(2)}$.

3. Choose $\mathcal{L}$ from Lemma B.1 corresponding to an approximation accuracy of $\delta_\epsilon^{(2)}$.

Now set

$$\mathcal{H} = \mathcal{P} \circ \overline{\mathcal{G}}_N \circ \mathcal{L}. \tag{28}$$

Let $f \in A$ be arbitrary. Then the triangle inequality gives

$$\|\mathcal{H}[f] - \mathcal{G}_N[f]\|_{L^2} \leq \|\mathcal{P}\overline{\mathcal{G}}_N\mathcal{L}[f] - \mathcal{P}\overline{\mathcal{G}}_N\mathcal{B}_N[f]\|_{L^2} \quad \text{(I)}$$
$$+ \|\mathcal{P}\overline{\mathcal{G}}_N(\mathcal{B}_N[f]) - \mathcal{P}\widehat{\mathcal{G}}_N(\mathcal{B}_N[f])\|_{L^2} \quad \text{(II)}$$
$$+ \|\mathcal{P}(\widehat{\mathcal{G}}_N\mathcal{B}_N[f]) - \mathcal{B}_N^{-1}(\widehat{\mathcal{G}}_N\mathcal{B}_N[f])\|_{L^2}. \quad \text{(III)}$$

We bound (I) by noting that the uniform approximation in Lemma B.1 gives, for any $x \in \Omega$,

$$\|\mathcal{L}[f](x) - \mathcal{B}_N[f](x)\|_{\ell^\infty(\mathbb{R}^N)} < \delta_\epsilon^{(2)},$$

so by continuity, (I) $\leq \epsilon/8$. In (II), for any $c \in \mathcal{B}_N\mathcal{T}_N(A)$,

$$\|\overline{\mathcal{G}}_N(c) - \widehat{\mathcal{G}}_N(c)\|_{\ell^\infty(\mathbb{R}^N)} < \delta_\epsilon^{(1)},$$

so (II) $\leq \epsilon/8$. Lastly, we have (III) $< \epsilon/4$ by construction. The result (19) immediately follows. $\square$

**Lemma B.1.** *Let $N \in \mathbb{N}$ be arbitrary. Let $A \subset (L^2(\Omega; \mathbb{R}), \|\cdot\|_{L^2(\Omega)})$ be compact. For any $\epsilon > 0$, there exists a continuous operator $\mathcal{L} : A \to L^2(\Omega; \mathbb{R}^N)$ such that*

$$\sup_{f \in A,\, x \in \Omega} \|\mathcal{L}[f](x) - \mathcal{B}_N[f](x)\|_{\ell^\infty(\mathbb{R}^N)} < \epsilon.$$

*Proof.* Let $A \subset (L^2(\Omega; \mathbb{R}), \|\cdot\|_{L^2(\Omega)})$ be compact. Let $N \in \mathbb{N}$ and $\epsilon > 0$ be arbitrary. Let $K(x,y)$ be a continuous positive-definite kernel on $\Omega$ with eigenfunctions $\psi_n(x)$. Observe that for any $f \in A$,

$$\int_\Omega \psi_n(x)\psi_n(y)f(y)\,\mathrm{d}y = \langle f, \psi_n \rangle \psi_n(x), \qquad n \in \mathbb{N}.$$

Note that the kernel $\tilde{K}_n(x,y) = \psi_n(x)\psi_n(y)$ is positive-definite. By defining $\tilde{\boldsymbol{K}}(x,y) = (\tilde{K}_n(x,y))_{n \in [N]}$, we have

$$\widehat{\mathcal{I}}^{(0)}[f] = \int_\Omega \tilde{\boldsymbol{K}}(x,y)\,f(y)\,\mathrm{d}y = (\langle f, \psi_n \rangle \psi_n)_{n \in [N]}$$

where $\widehat{\mathcal{I}}^{(0)} : A \to A^N$. Now define the affine operator $\widehat{\mathcal{I}}^{(1)} : A^N \to A$ by $\widehat{\mathcal{I}}^{(1)}[f](x) = \boldsymbol{1}_N(f(x))$, which gives

$$(\widehat{\mathcal{I}}^{(1)} \circ \widehat{\mathcal{I}}^{(0)})[f](x) = \sum_{n \in [N]} \langle f, \psi_n \rangle \psi_n(x) = \mathcal{T}_N[f](x)$$

for all $f \in A$.

Next, consider the mapping $\tilde{h} : \mathbb{R}^{d+1} \to \mathbb{R}^{N+1}$ defined as $\tilde{h}(a, x) = (a, \psi_1(x), \ldots, \psi_N(x))$. We wish to restrict $\tilde{h}$ to a compact domain and apply the universal approximation theorem for MLPs. To do that, we need to bound

$$\sup_{f \in A} \|\mathcal{T}_N[f]\|_{L^\infty} .$$

Since $K(x, y)$ is continuous, then $\psi_n$ is continuous for all $n \in \mathbb{N}$, and since $\Omega$ is compact, then each $\psi_n$ attains some finite maximum $M_n < \infty$ on $\Omega$. As a result, for each $n \in [N]$, we have $\|\psi_n\|_{L^\infty} \leq Q := \max_{n \in [N]} M_n$. Furthermore, for any $f \in A$, Hölder's inequality gives

$$\sum_{n \in [N]} |\langle f, \psi_n \rangle| \leq \left( \sum_{n \in [N]} |\langle f, \psi_n \rangle|^2 \right)^{1/2} \sqrt{N} = \sqrt{N} \|\mathcal{T}_N[f]\|_{L^2}$$

since $\langle \psi_n, \psi_m \rangle = \delta_{nm}$. Since $A$ is compact, then there exists $C > 0$ such that for any $f \in A$, $\|f\|_{L^2} \leq C$. But $\|\mathcal{T}_N[f]\|_{L^2} \leq \|f\|_{L^2}$, so

$$\|\mathcal{T}_N[f]\|_{L^\infty} \leq \sum_{n \in [N]} |\langle f, \psi_n \rangle| \, \|\psi_n\|_{L^\infty} \leq QC\sqrt{N} := \tilde{C} .$$

So now take $h : [-\tilde{C}, \tilde{C}] \times \Omega \to \mathbb{R}^N$ defined by $\tilde{h}(a, x) = (a\psi_1(x), \ldots, a\psi_N(x))$. Since each $\psi_n$ is continuous, then $h$ is a continuous function on a compact set, so there exists an MLP $\mathcal{N}^{\text{lift}} : [-\tilde{C}, \tilde{C}] \times \Omega \to \mathbb{R}^N$ such that

$$\sup_{\substack{a \in [-C, C] \\ x \in \Omega}} \|\mathcal{N}^{\text{lift}}(a, x) - (a\psi_1(x), \ldots, a\psi_N(x))\|_{\ell^\infty(\mathbb{R}^N)} < \epsilon/\mu(\Omega) ,$$

where $\mu$ is the standard Lebesgue measure. Set

$$\mathcal{L}[f](x) = \int_\Omega \mathcal{N}^{\text{lift}} \left( (\widehat{\mathcal{I}}^{(1)} \circ \widehat{\mathcal{I}}^{(0)})[f](y), y \right) \, \mathrm{d}y .$$

Since $\mathcal{N}$ is uniformly continuous, then $\widetilde{\mathcal{N}}$ is continuous, and since integration is continuous, then $\mathcal{L}$ is continuous. For any $f \in A$, $x \in \Omega$, and $n \in [N]$, we have

$$
\begin{aligned}
|(\mathcal{L}[f](x))_n - \langle f, \psi_n \rangle| &= \left| \int_\Omega \mathcal{N}^{\text{lift}}(\mathcal{T}_N[f](y), y)_n \, \mathrm{d}y - \int_\Omega f(y) \psi_n(y) \, \mathrm{d}y \right| \\
&\leq \int_\Omega \left| \mathcal{N}^{\text{lift}}(T_N[f](y), y)_n - T_N[f](y)\psi_n(y) \right| \, \mathrm{d}y \\
&< \epsilon ,
\end{aligned}
$$

which completes the proof. $\qquad \square$

**Lemma B.2.** *Let $N \in \mathbb{N}$ be arbitrary. Let $\{\psi_n\}_{n \in \mathbb{N}}$ be a continuous and orthonormal basis for $L^2(\Omega; \mathbb{R})$, and let $A \subset (L^2(\Omega; \mathbb{R}), \|\cdot\|_{L^2})$ be compact. Define $S = span\{\psi_n\}_{n \in [N]} \cap A$. For any $\epsilon > 0$, there exists a continuous operator $\mathcal{P} : \mathbb{R}^N \to L^2(\Omega; \mathbb{R})$ such that, for all $f \in S$,*

$$\|\mathcal{P}(\langle f, \psi_1 \rangle, \ldots, \langle f, \psi_N \rangle) - f\|_{L^2(\Omega)} < \epsilon .$$

*Proof.* Let $N \in \mathbb{N}$ and $\epsilon > 0$ be arbitrary. For any $f \in S$, we have

$$\sum_{n \in [N]} |\langle f, \psi_n \rangle| \leq \left( \sum_{n \in [N]} |\langle f, \psi_n \rangle|^2 \right)^{1/2} \sqrt{N} = \sqrt{N} \|f\|_{L^2}$$

by Hölder's inequality. Since $A$ is compact, then there exists $C > 0$ such that for any $f \in S$, $\|f\|_{L^2} < C$. Thus, for any $f \in S$ and $n \in [N]$,

$$|\langle f, \psi_n \rangle| \leq C\sqrt{N}$$

Consider the mapping $h : [-C\sqrt{N}, C\sqrt{N}]^N \times \Omega \to \mathbb{R}^N$ defined by

$$h(a_1, \ldots, a_N, x) = (a_1\psi_1(x), \ldots, a_N\psi_N(x)) \,.$$

Since each $\psi_n$ is continuous, then $h$ is continuous and defined on a compact set. So by the universal approximation theorem for MLPs, there exists an MLP $\mathcal{N}^{\text{proj}}$ such that

$$\sup_{a \in [C\sqrt{N}, C\sqrt{N}]^N, \, x \in \Omega} \|\mathcal{N}^{\text{proj}}(a_1, \ldots, a_N, x) - h(a_1, \ldots, a_N, x)\|_{\ell^\infty(\mathbb{R}^N)} < \epsilon / \left(N\sqrt{\mu(\Omega)}\right) \,,$$

where $\mu$ is the standard Lebesgue measure. Define the pointwise operator $\mathcal{I} : L^2(\Omega; \mathbb{R}^N) \to L^2(\Omega; \mathbb{R})$ as $\mathcal{I}[f](x) = \mathbf{1}_N^\top(f(x))$. Set $\mathcal{P}[a_1, \ldots, a_N](x) = \mathcal{I} \circ \mathcal{N}^{\text{proj}}(a_1, \ldots, a_N, x)$. For any $f \in S$,

$$\|\mathcal{P}(\mathcal{B}[f]) - f\|_{L^2}^2 = \int_\Omega \left( \sum_{n \in [N]} \mathcal{N}^{\text{proj}}(\mathcal{B}[f], y)_n - \langle f, \psi_n \rangle \psi_n(y) \right)^2 \mathrm{d}x$$

$$< \int_\Omega \left( \sum_{n \in [N]} \frac{\epsilon}{N\sqrt{\mu(\Omega)}} \right)^2 \mathrm{d}y$$

$$= \epsilon^2 \,,$$

from which the result immediately follows. □

## B.2 Finite-Dimensional Case

The following is a restatement and proof of Theorem 3.2.

**Theorem 3.2.** *Adopt the same assumptions as Theorem 3.1, but with $A' \subset C^1(\Omega; \mathbb{R})$, compact with respect to the $\|\cdot\|_{L^\infty}$ norm and with uniformly bounded first derivatives. Additionally, let $\{w^{(M)}\}_{M \in \mathbb{N}}$ and $X_M = \{x^{(M)}\}_{M \in \mathbb{N}}$ define a sequence of $M$-point quadrature rules on $\Omega$. Suppose that there exists $C > 0$ such that, for any $f \in C^1(\Omega; \mathbb{R})$,*

$$\left| \sum_{m \in [M]} w_m^{(M)} f(x_m^{(M)}) - \int_\Omega f(x)\, dx \right| \leq \frac{C\|\nabla f\|_{L^\infty}}{M} \,.$$

*For any $\epsilon > 0$, there exists $M \in \mathbb{N}$, $\nu > 0$, and $\widetilde{\mathcal{H}}_M : \mathbb{R}^{N_T} \to \mathbb{R}^{N_T}$ of the form* (14) *such that*

$$\sup_{f \in A'} \left\| \widetilde{\mathcal{H}}_M(\boldsymbol{f}_T) - (\mathcal{G}[f])\big|_{X_T} \right\|_{\ell^\infty(\mathbb{R}^M)} < \epsilon + \nu\, h_{\Omega, X_T} \,, \tag{29}$$

*where $\boldsymbol{f}_T = \{f(x)\}_{x \in X_T}$,*

$$h_{\Omega, X_T} = \sup_{x \in \Omega} \min_{x_i \in X_T} \|x - x_i\|$$

*is the fill distance, and $\widetilde{\mathcal{H}}_M$ depends parametrically on the quadrature nodes $X_M = \{x_m^{(M)}\}_{m \in [M]}$.*

*Proof.* Let $\widetilde{K}(x, y)$ be a positive-definite kernel used to interpolate values from $X_T$ to $X_M$. For any $M \in \mathbb{N}$, $\widetilde{\mathcal{H}}_M : \mathbb{R}^M \to \mathbb{R}$, $\mathcal{H} : A' \to C^1(\Omega; \mathbb{R})$, and $f \in A'$, we have

$$\|\widetilde{\mathcal{H}}_M(\boldsymbol{f}_{X_T}) - \mathcal{G}[f](X_T)\|_{\ell^\infty} \leq \|\widetilde{\mathcal{H}}_M(\boldsymbol{f}_{X_T}) - \mathcal{H}[f](X_T)\|_{\ell^\infty} + \|\mathcal{H}[f](X_T) - \mathcal{G}[f](X_T)\|_{\ell^\infty}$$

Note that $A' \subset L^2(\Omega; \mathbb{R})$ and is compact with respect to $\|\cdot\|_{L^2(\Omega)}$, so Theorem 3.1 applies. There exists $N \in \mathbb{N}$ and

$$\mathcal{H} = \mathcal{P} \circ \overline{\mathcal{G}}_N \circ \mathcal{L}$$

as defined by (28) such that

$$\sup_{f \in A'} \|\mathcal{H}[f] - \mathcal{G}[f]\|_{L^2(\Omega)} < \epsilon/2 \,.$$

Since $A'$ is compact and $\mathcal{H}$ and $\mathcal{G}$ are continuous (by construction and by assumption, respectively), then the image $(\mathcal{H} - \mathcal{G})(A')$ is compact, so there exists a constant $\hat{\nu}$ such that, for all $f \in A'$,

$$\|\mathcal{H}[f](X_T) - \mathcal{G}[f](X_T)\|_{\ell^\infty(\mathbb{R}^{N_T})} \leq \hat{\nu}\, h_{\Omega, X_T} + \epsilon/2\,. \tag{30}$$

Consider $\mathcal{H}$. The projector $\mathcal{P}$ is already an MLP, and $\overline{\mathcal{G}}_N$ can be expressed as (17) with $\boldsymbol{K}^{(\ell)} \equiv 0$. It remains to construct an MLP lifting operator $\overline{\mathcal{L}}$.

Since $A'$ is compact, there exists $\beta > 0$ such that, for all $f \in A'$, $\|\nabla f\|_{L^\infty} < \beta$. Therefore, for all $f \in A'$,

$$\left| \sum_{m \in [M]} w_m^{(M)} f(x_m^{(M)}) - \int_\Omega f(x)\, \mathrm{d}x \right| \leq \frac{C\beta}{M}\,,$$

where the constants are independent of $f$. Moreover, any continuous map $\mathcal{F}$ of $A'$ has a similar bound, which depends on $\mathcal{F}$.

Consider $\mathcal{N}^{\text{lift}}$, $\widehat{\mathcal{I}}^{(1)}$ and $\widehat{\mathcal{I}}^{(0)}$ as defined in the proof of Lemma B.1. We may assume, without loss of generality, that $\sigma(x)$ is Lipschitz continuous, which makes $\mathcal{N}^{\text{lift}}$ Lipschitz continuous. Let $\tilde{\epsilon} > 0$ be arbitrary and $\mu$ denote the standard Lebesgue measure. Since $\mathcal{N}^{\text{lift}}$ is Lipschitz, there exists $\gamma_L > 0$ such that $\|\mathcal{N}(a, x) - \mathcal{N}(b, x)\|_{\ell^\infty} < \gamma_L |a - b|$. We can also write $\mathcal{N}^{\text{lift}}$ as

$$\mathcal{N}^{\text{lift}}(a, x) = \boldsymbol{W}^{(1)} \sigma \left( \boldsymbol{W}^{(0)} \begin{bmatrix} a \\ x \end{bmatrix} + \boldsymbol{b}^{(0)} \right)$$

for some $\boldsymbol{W}^{(0)} \in \mathbb{R}^{\tilde{p} \times 2}$, $\boldsymbol{W}^{(1)} \in \mathbb{R}^{N \times \tilde{p}}$, and $\boldsymbol{b} \in \mathbb{R}^{\tilde{p}}$. Furthermore, note that $\max_{n \in [N]} \|\psi_n\| := Q < \infty$, where $\psi_n$ are the Mercer eigenfunctions of $\widetilde{K}(x, y)$ and $N$ is the truncation level (i.e., the channel dimension). Next, define the kernel interpolant of $f$, denoted $\tilde{f}$, as

$$\tilde{f}(x) = \sum_{j \in [N_T]} \alpha_j \widetilde{K}(x, x_j), \qquad x_j \in X_T,$$

provided that the matrix

$$\boldsymbol{A}_{ij} = \widetilde{K}(x_i, x_j), \qquad x_i, x_j \in X_T$$

is invertible. From (Wendland, 2005, Thm. 11.13), and the compactness of $A'$, there exists $\tilde{\nu} > 0$ such that for all $f \in A'$

$$\|\tilde{f} - f\| < \tilde{\nu}\, h_{\Omega, X_T}\,.$$

since $A'$ is compact. There also exists $M_1$ (which is independent of $\tilde{f}$ but can depend on $\{\psi_n\}_{n \in [N]}$) such that, for all $M \geq M_1$ and $f \in A'$,

$$\max_{n \in [N]} \left| \sum_{m \in [M]} w_m f(x_m) \psi_n(x_m) - \int_\Omega f(x) \psi_n(x)\, \mathrm{d}x \right| < \frac{\tilde{\epsilon}}{N Q \gamma_L\, \mu(\Omega)}\,.$$

Defining

$$\tilde{c}_n := \int_\Omega \tilde{f}(x) \psi_n(x)\, \mathrm{d}x, \qquad \tilde{d}_n := \sum_{m \in [M]} w_m \tilde{f}(x_m) \psi_n(x_m)$$

gives, for any $x \in \Omega$,

$$\left| \sum_{n \in [N]} \tilde{c}_n \psi_n(x) - \sum_{n \in [N]} \tilde{d}_n \psi_n(x) \right| \leq \sum_{n \in [N]} (|\tilde{c}_n - \tilde{d}_n|) \|\psi_n\|_\infty < \frac{\tilde{\epsilon}}{\gamma_L\, \mu(\Omega)}\,.$$

Now, define the matrices

$$\boldsymbol{K} = \begin{bmatrix} \boldsymbol{K}^{(1)} \\ \vdots \\ \boldsymbol{K}^{(N)} \end{bmatrix}, \qquad \boldsymbol{K}_{ij}^{(n)} = w_j^{(M)}\psi_n(x_i^{(M)})\psi_n(x_j^{(M)}), \qquad i,j \in [M],$$

and set

$$\overline{\mathcal{L}}_M(\boldsymbol{f}_T) = \boldsymbol{W}^{(1)}\sigma\left(\boldsymbol{W}^{(0)}\begin{bmatrix} \mathbf{1}_N^\top \, \texttt{reshape}(\boldsymbol{K}\,\widetilde{\boldsymbol{K}}_{M,T}\widetilde{\boldsymbol{K}}_{T,T}^{-1}\boldsymbol{f}_T, (N,M)) \\ X_M^\top \end{bmatrix} + \boldsymbol{b}^{(0)}\right)\boldsymbol{w}^{(M)} \tag{31}$$

where $\sigma(x)$ is Lipschitz continuous and non-polynomial, all vectors are understood as column vectors, and

$$(\widetilde{\boldsymbol{K}}_{*,\dagger})_{ij} = \widetilde{K}(x_i^{(*)}, x_j^{(\dagger)}), \qquad \boldsymbol{w}^{(M)} = (w_1^{(M)}, \ldots, w_M^{(M)}).$$

Unpacking (31), the matvec $\widetilde{\boldsymbol{K}}_{M,T}\widetilde{\boldsymbol{K}}_{T,T}^{-1}\boldsymbol{f}_T$ maps the function values on $X_T$ to the kernel interpolant evaluated on $X_M$. The matrix $\boldsymbol{K}$ approximates the integral

$$\int_\Omega \psi_n(x)\psi_n(y)\tilde{f}(y)\,\mathrm{d}y\,,$$

and the reshape and multiplication by $\mathbf{1}_N^\top$ sums over the $N$ basis functions to yield

$$\sum_{n\in[N]} \tilde{d}_n\psi_n(x_m^{(M)})\,.$$

The final multiplication by $\boldsymbol{w}^{(M)}$ approximates the integral

$$\int_\Omega \sum_{n\in[N]} \tilde{d}_n\psi_n(x)\psi_m(x)\,\mathrm{d}x \approx \tilde{d}_m.$$

Lastly, by assumption, the quadrature rule exactly integrates constant functions, so by taking $M \geq M_1$, we obtain

$$\begin{aligned}
\|\overline{\mathcal{L}}_M(\boldsymbol{f}_T) - (\mathcal{L}[f])\big|_{X_T}\|_{\ell^\infty(\mathbb{R}^N)} &\leq \|\overline{\mathcal{L}}_M(\boldsymbol{f}_T) - \overline{\mathcal{L}}_M(\boldsymbol{f}_M)\|_{\ell^\infty(\mathbb{R}^N)} \\
&\quad + \|\overline{\mathcal{L}}_M(\boldsymbol{f}_M) - (\mathcal{L}[f])\big|_{X_M}\|_{\ell^\infty(\mathbb{R}^N)} \\
&\leq \gamma_L\tilde{\nu}\,h_{\Omega,X_T} + \tilde{\epsilon}\,,
\end{aligned}$$

where $\boldsymbol{f}_M$ uses $X_T = X_M$.

To build $\widetilde{\mathcal{H}}_M$, we set

$$\widetilde{\mathcal{H}}_M = \mathcal{P}_{X_T} \circ \overline{\mathcal{G}} \circ \overline{\mathcal{L}}_M$$

where $\mathcal{P}_{X_T}$ is the projection from channel space onto $X_T$. We then note that, without loss of generality, $\mathcal{P}_{X_T}$ and $\overline{\mathcal{G}}$ are Lipschitz continuous, with constants $\gamma_P$ and $\gamma_G$. Taking $\tilde{\epsilon} = \epsilon/(2\gamma_P\gamma_G)$ and setting $\nu = \gamma_P\gamma_G\gamma_L\tilde{\nu} + \hat{\nu}$ [cf. (30)] completes the proof. $\qquad\square$

