# OpenReview forum: "Kernel Neural Operators (KNOs) for Scalable, Memory-efficient, Geometrically-flexible Operator Learning"
_TMLR — Accepted by TMLR_

### Review · Reviewer_VpVj · 2026-02-05

**Summary Of Contributions:**

This paper introduces Kernel Neural Operators (KNOs), a kernel-based neural operator architecture that explicitly parameterizes integral kernels and decouples kernel choice from numerical discretization via quadrature. This design enables geometrically flexible operator learning on regular grids, irregular domains, meshes, and point clouds, while supporting highly expressive non-stationary and anisotropic kernels. The authors provide universal approximation theorems for both continuous and discretized KNOs, establishing theoretical soundness. Empirically, KNOs achieve comparable or superior accuracy to state-of-the-art neural operators such as FNOs, GNOTs, and Transolvers across a range of PDE benchmarks, while using one to two orders of magnitude fewer trainable parameters, resulting in substantially reduced memory requirements

**Additional Comments:**

1. In Equation (9), for each j, it may be clearer to use a scalar y_j​ instead of the vector y to avoid confusion and better distinguish it from the full vector notation.

2. Did the authors evaluate the model using different training and testing grids?

3. Why is the kernel trained on regular grids, even though interpolation is required for irregular grids? Is this related to the use of the FFT? If so, it would be helpful to present the complete model structure, including the role of the FFT.

**Audience:**

Yes

**Audience Explanation:**

The combination of explicit kernels, numerical quadrature, and deep learning is novel and aligns well with TMLR’s emphasis on principled ML methods. The demonstrated memory efficiency is particularly relevant for researchers concerned with deployment, scalability, and hardware constraints.

**Broader Impact Concerns:**

No.

**Claims And Evidence:**

No

**Claims Explanation:**

Accuracy and parameter efficiency claims are not well supported by quantitative comparisons (Tables 2 and 4). Although KNO uses fewer parameters across all benchmarks, it does not always achieve the best performance. This may be due to insufficient model capacity. Additional experiments with larger parameter budgets are needed to better verify the approximation capability of KNO.

**Requested Changes:**

1. There exist several neural network models that claim to achieve comparable performance with reduced parameter counts or lower computational cost, including Sharma & Shankar (2025) on ensemble and mixture-of-experts DeepONets (which is mentioned by the authors but not evaluated experimentally), BelNet (Zhang et al., 2023), GIT-Net (Wang et al., 2024), and Integral Neural Networks (Solodskikh et al., 2023). All of these methods are based on integral representations and emphasize reduced computational or memory requirements. It would be beneficial for the authors to discuss these related works, or other models with similar claims, and to further analyze the advantages of KNO through additional experiments or a focused comparative discussion.

2. The authors claim that KNO achieves comparable accuracy with significantly fewer parameters, which is illustrated by the experimental results in Tables 2 and 4. However, these results alone are not sufficient to demonstrate that KNO is more efficient than other models. Although KNO consistently uses fewer parameters, it achieves the best performance in only two benchmark cases (Table 2). In the Beijing-Air case, this improvement may be attributed to the fact that KNO uses a parameter count similar to that of FNO. To more convincingly demonstrate comparable approximation capability and ensure a fair comparison, additional experiments are needed in which KNO attains the same or better accuracy while using a similar or smaller number of parameters. Notably, larger KNO models are explored in the ablation studies; therefore, it would be valuable to directly compare results where KNO matches or exceeds the accuracy of other models under comparable parameter budgets.

3. Some sections (e.g., Section 2.2 on kernels) are quite dense and would benefit from a more intuitive summary or schematic. The authors could also provide a clearer illustration of the overall model structure.

4. Typos and minor grammatical issues are rare but present and should be corrected in a final pass.

5. The results show that certain kernels perform better in specific cases; however, the underlying reasons are not discussed. It would be helpful for the authors to analyze these differences, as they may be related to the properties of the underlying physical models.

---

> ### Author Response · Authors · 2026-03-04
> **Response to Reviewer VpVj**
>
> We thank the reviewer for their very careful read of the work and their comments. Our responses to requested changes are below:
>
> 1) Re ensemble models (SharmaShankar), BelNet, Integral Neural Networks (INNs), GitNet: we thank the reviewer for pointing these out. We will add details about ensemble models and BelNet to the introduction. We did reference and discuss the INN work in Section 1.1; we will edit the paper to add that the INN work was formulated explicitly for image tasks on regular grids, while the KNO is more general. As for the comparison with ensemble models, we don’t believe a direct comparison is exactly fair since the KNO is not an ensemble model. That said, our results are uniformly more accurate than those of Sharma and Shankar 2025; we will edit the text to mention this. GitNet also seems like a fascinating architecture that leverages learning bases that diagonalize the operator. We will add a discussion of the GitNet architecture to the introduction, though we do believe the KNO is more directly comparable to the FNO family of neural operators. It may be possible in future work to incorporate elements of the GitNet architecture into the KNO, and vice-versa. Finally, we were aware of BelNet but could not find an implementation. The KNO can be thought of as a nonlinear deep generalization of BelNets when certain specific choices are made, but BelNets require using neural networks as eigenfunctions to a kernel (thereby inflating the parameter count) while the KNO directly uses closed-form kernels. We will add this discussion to the introduction as well.

---

> > ### Author Response · Authors · 2026-03-04
> > **Response to Reviewer VpVj continued**
> >
> > 2) Re parameter counts and accuracy, the reviewer is making an excellent point here. We explored this in response to the reviewer’s comment and made two observations. For one, we mistakenly used a different test set on the NS-Pipe problem than was used for the other methods, simply because we indexed the functions directly after the training set instead of indexing from the end. We report an updated result here for a KNO using a ns-gsm kernel, a depth of 7 and a channel dimension of 128, in which KNO maintains its relative parameter efficiency and bests the other models.
> >
> >     ### NS-Pipe
> >
> >     | Model | KNO | Transolver (2nd best)|
> >     |-|-|-|
> >     | % NRMSE | 0.317 ± 0.0150 | 0.33 ± 0.02 |
> >     | Param count | 274,561 |2,810,817 |
> >
> >     We also performed an ablation study over model size on this dataset to compliment those in A.7.3. In this study the fixed model had a ns-gsm kernel, a depth of 5 and a channel dimension of 64. Here one is able to achieve a more performant KNO by increasing depth or width, but notice depth adds expressivity with a smaller parameter cost.
> >
> >
> >     | **Depth** |   4  |  5 |  6  |  7  |  8  |
> >     |---------------------------|---------|---------|---------|---------|----------|
> >     | % NRMSE   | 0.434 ± 0.0147  | 0.386 ± 0.025 | 0.401 ± 0.0210  | 0.352 ± 0.004 |  0.348 ± 0.021 |
> >     | Param count | 61,057 | 74,177 | 87,297 | 100,417 | 113,537 |
> >
> >     | **Channels** |   16  |  32 |  64  |  128  |  192  | 256 |
> >     |---------------------------|---------|---------|---------|---------|----------|----|
> >     | % NRMSE   | 1.031 ± 0.018  | 0.579 ± 0.006 | 0.386 ± 0.025  | 0.360 ± 0.005 |  0.350 ± 0.007 | 0.343 ± 0.006 |
> >     | Param count | 13,169 | 29,921 | 74,177 | 205,697 | 394,561 | 640,769 |
> >
> >
> >     Next, for the NS Mach 1.0 problem, KNO was able to perform much better after changing the normalization strategy to match Transolver's: from pointwise (per spatial location) normalization across training samples to global normalization using a single mean and standard deviation computed over the entire training set. So that the relative improvement of Transolver's result becomes 1.05%.
> >
> >     ### NS Mach 1.0
> >
> >     | **Model** | KNO| Transolver |
> >     |-|-|-|
> >     | % NRMSE | 48.636 ± 0.073 | 48.127 ± 0.483 |
> >     | Param count | 31,105 | 3,791,377 |
> >
> >     ### Amended Results
> >
> >     The amended results including standard errors (Table 8) are below, which we hope alleviate the reviewers concern about KNO's expressivity given that in these updates KNO is the second or first best model in all cases. For Burgers', KNO is worse than solely FNO. For Beijing-air, Darcy (PWC) and NS-Pipe, KNO is the best performing model. For Darcy (triangle), the KM does the best, but KNO matches FNO with regard to standard error. For NS Mach 1.0, KNO performs 1.05% worse than Transolver with two orders of magnitude less parameters, but better than all other methods. In React.-Diff. KNO outperforms all operator learning methods except KM. Overall, KNO performs better than all of FNO, GNOT and Transolver in five of seven benchmark problems.
> >
> >
> >     | Problem              | FNO                | GNOT               | Transolver           | KM       | KNO                |
> >     |----------------------|--------------------|--------------------|----------------------|----------|--------------------|
> >     | Burgers'     | **0.276 ± 0.004**  | 0.89 ± 0.014       | 1.077 ± 0.101        | 2.831    | 0.574 ± 0.011      |
> >     | Beijing-Air          | 55.33 ± 0.18       | 40.3 ± 0.21        | 41.68 ± 0.332        | 50.982   | **24.941 ± 0.161** |
> >     | Darcy (PWC)          | 1.79 ± 0.025       | 2.58 ± 0.034       | 1.989 ± 0.0278       | 3.064    | **1.512 ± 0.012**  |
> >     | Darcy (triangle)     | 0.043 ± 0.003  | 0.111 ± 0.013      | —                    | **0.033**| 0.045 ± 0.002      |
> >     | NS-Pipe              | 0.67               | 0.47 ± 0.02        | 0.33 ± 0.02      | 2.742    | **0.317 ± 0.015** |
> >     | NS Mach 1.0          | 58.05 ± 4.77       | 81.5 ± 2.18        | **48.127 ± 0.483**   | 54.150   | 48.636 ± 0.073     |
> >     | React.-Diff.         | 6.68e-3 ± 2.77e-4  | 4.47e-3 ± 9.45e-5  | 8.10e-3 ± 1.89e-5    | **8.75e-05** | 9.20e-4 ± 1.02e-4 |
> >
> >     Finally, we mention the several ablation studies in the appendix that we performed to test larger KNOs, in Table 13 and Table 14 ablating depth/width on Darcy (PWC) and Beijing-Air respectively and in Table 12 ablating over larger KNOs by way of the MVK structure for Burgers' and Darcy (PWC).

---

> > > ### Author Response · Authors · 2026-03-04
> > > **Response to Reviewer VpVj continued**
> > >
> > > 3) We apologize for the density of text. We will add schematics to Section 2.2 and a figure of the overall model to make the writing more intuitive.
> > >
> > > 4) We apologize for these also and will fix these on revision.
> > >
> > > 5) Re when certain kernels perform better: we apologize for the lack of clarity in the text. We observed that the more expressive kernels (NS-GSM) always performed better except on very noisy datasets (Beijing-Air due to it being real world, NS Mach 1 due to turbulence), where they were more difficult to train than the GSM kernels. We will edit the text to better clarify this.
> > >
> > > We also address the additional comments below:
> > >
> > > 1) Re equation (9), agreed, this is quite reasonable.
> > >
> > > 2) We did, but not comprehensively. We show a super-resolution result in Appendix A.1.5 where we tested on a finer grid and observed the same errors.
> > >
> > > 3) Good question. We do not use the FFT for our work in general. We used regular (tensor-product) grids on tensor-product domains (as opposed to irregular domains) because the regular grids allow for a highly-efficient dimension-wise factorization of our integral operators, greatly speeding up training and inference. On irregular domains, we used irregular “grids” (meshes and even point clouds) since tensor-product grids do not generally map neatly to irregular domains. We point this out in the text, but perhaps could have done a better job!

---

> > > > ### Comment · Reviewer_VpVj · 2026-03-25
> > > > **Reference**
> > > >
> > > > Some references have published versions more than arXiv, please use the official ones. I found the following:
> > > >
> > > > Transolver: A fast transformer solver for PDEs on general geometries, 2024, ICML
> > > >
> > > > PDEBENCH: An extensive benchmark for scientific machine learning, 2024, ICLR
> > > >
> > > > Neural non-stationary spectral kernel, 2018, NeurIPS
> > > >
> > > > Please also find others.

---

> > > > > ### Author Response · Authors · 2026-03-25
> > > > >
> > > > > Thank you for pointing this out. We will update the list of references.

---

### Review · Reviewer_gR1i · 2026-02-08

**Summary Of Contributions:**

Based on the architecture of Fourier Neural Operator (FNO), this paper introduces Kernel Neural Operator (KNO), which is built from compositions of kernel-based integral operators with nonlinear activations, lifting and projection maps between input and output function spaces. The main contribution of the paper is to explicitly parameterize the kernels of the integral operators via neural networks, which is more expressive than standard FNO and can be applied to different geometries (regular grids, triangular meshes, point clouds, etc) in a more flexible way. Both theoretical analysis and numerical experiments are provided to justify the effectiveness of the proposed methodology.

**Additional Comments:**

References:

[1] Li, Zongyi, et al. "Physics-informed neural operator for learning partial differential equations." ACM/IMS Journal of Data Science 1.3 (2024): 1-27.

[2] Khoo, Yuehaw, Jianfeng Lu, and Lexing Ying. "Solving parametric PDE problems with artificial neural networks." European Journal of Applied Mathematics 32.3 (2021): 421-435.

[3] Fan, Yuwei, Cindy Orozco Bohorquez, and Lexing Ying. "BCR-Net: A neural network based on the nonstandard wavelet form." Journal of Computational Physics 384 (2019): 1-15.

[4] Feliu-Faba, Jordi, Yuwei Fan, and Lexing Ying. "Meta-learning pseudo-differential operators with deep neural networks." Journal of computational physics 408 (2020): 109309.

[5] Long, Zichao, et al. "Pde-net: Learning pdes from data." International conference on machine learning. PMLR, 2018.

[6] Long, Zichao, Yiping Lu, and Bin Dong. "PDE-Net 2.0: Learning PDEs from data with a numeric-symbolic hybrid deep network." Journal of Computational Physics 399 (2019): 108925.

**Audience:**

Yes

**Audience Explanation:**

To the best of the reviewer's knowledge, AI4Science and Scientific Machine Learning (in particular, solving PDEs via machine learning based methods like physics-informed neural networks or operator learning) have been a hot research topic nowadays. Given that the proposed method is both theoretically grounded and validated on a wide range of PDE examples, the reviewer thinks that TMLR's audience will be interested in the findings of the paper.

**Broader Impact Concerns:**

Since this paper is mainly about scientific machine learning and operator learning, the reviewer doesn't think there is any related concern on the work's ethical implications.

**Claims And Evidence:**

Yes

**Claims Explanation:**

The reviewer thinks that the results presented in the paper are supported by accurate and clear evidence. On the one hand, the theoretical results (universal approximation theorems in Theorem 3.1 and 3.2) are clearly stated with proofs in Appendix B. On the other hand, details of the experiments used for supporting the empirical claims, such as timings, parameter counts and statistical significance, are all provided in both the main text and the appendix of the paper.

**Requested Changes:**

The reviewer's first main concern is that the paper's novelty seems to be incremental compared to existing frameworks in operator learning. Specifically, the main difference between KNO and FNO is that the kernels in the integral operator are parametrized in a "learnable" way, which makes the architecture more expressive. However, existing work like [3,4] based on pseudo-differential operators have already shown that the matrices associated with the integral operators are low rank for most cases, so it might be sufficient to use the diagonal matrices adopted in FNO? The authors might need to further clarify on the contribution part in the revision of the paper.

Moreover, it seems to the reviewer that the literature review on existing work and baselines seem to be lacking. For instance, there has been a few related work prior to operator learning from the community, which the authors might consider adding to the references [2,3,4,5,6]. Also, would it be possible for the authors to compare the proposed method with other variants of FNO, such as PINO [1] and others? It would be interesting to investigate how a physics-informed loss might help with the training of KNO as well.

---

> ### Author Response · Authors · 2026-03-04
> **Response to Reviewer gR1i**
>
> We appreciate the reviewer’s response. We respond to the “requested changes” here:
>
> 1) Regarding the issue of the KNO being incremental: We can certainly see the reviewer’s perspective, but would respectfully like to counter. The KNO makes two major changes: (1) allowing for closed-form kernels makes the KNO achieve similar errors with far fewer trainable parameters, which we genuinely believe is an interesting and non-intuitive result; (2) the introduction of quadrature allows us to naturally handle both irregular and regular domains, while other neural operators need specialized machinery to do so ( a mapping network like in the Geo-FNO). We believe these two changes better align neural operators with scientific computing workflows that generate the datasets for neural operators in the first place.
>
> 2) In the same point, the reviewer also mentions the issue of diagonal and low-rank matrices. We agree that diagonal matrices are perfectly reasonable. In fact, we made this choice ourselves in this work; all our results in the main paper are for diagonal matrix-valued kernels (in the channel dimension, like the FNO). We will edit the text to make this clearer. Interestingly though, the KNO allows for other choices of matrix-valued kernels, which the FNO cannot without serious modification. For instance in Appendix A.7.2, we found that a symmetric banded matrix-valued kernel improved performance on two of the benchmark problems.
>
> 3) Regarding the comment on the literature review, we sincerely apologize for the oversight. We will include these references into our introduction and related work sections.
>
> 4) Regarding physics-informed losses: we agree that a physics-informed loss is the next reasonable step for the KNO. This paper was focused on creating the baseline KNO and thoroughly exploring its parameter counts and architectural choices, and comparing against the most similar baselines. We anticipate that a paper with physics-informed losses would indeed need comparisons against the PINO and other variants, all of which appear very promising.

---

### Review · Reviewer_nzrj · 2026-02-20

**Summary Of Contributions:**

In this work, the authors introduce Kernel Neural Operators (KNOs), a novel architecture for doing operator learning. The main novelty of KNOs, are that they allow learnable kernels of the integral transformations, as opposed to, e.g. FNOs, that use a fixed kernel. This allows KNOs to be used on irregular geometries, among other benefits. The authors produce both theoretical and applied results, showing that KNOs are universal approximators, and compare KNOs to other state-of-the-art-approaches on various problems.

**Audience:**

No

**Audience Explanation:**

The KNO is a new model with many interesting components, but I don't think any of them are interesting, significant, or novel enough to make an impact. The manuscript opens with a strong promise of a new architecture with many significant contributions (generalized architecture, strong theory, apples to irregular geometries, ameliorates the curse of dimensionality, expressive non-trivial kernels, suite of numerical results showing strong performance, reduced memory size) but a close read of the work shows that each contribution, in my opinion, is order epsilon. A manuscript body that follows through on the promises made in the abstract could be outstanding, however I don't think this body does. A point-by-point criticism is given below
 - _Generalized architecture_ identifying new architectures is important, especially if they outperform, unify, or simplify previous architectures. I'm not sure, however, if the KNO architecture does any of these things.
 - _Theoretical Contributions_ Two theorems constitute the theoretical contributions, Theorem 3.1, and 3.2. The proofs of the theorems are correct, but the novelty seems very low to me. Theorem 3.1 is a totally standard proof to me (in short, truncate an infinite-dimensional operator and use invoke standard MLP universality). Theorem 3.2 is better, but again has limited novelty. Using a consistent integral quadrature maintains universality. Where's the insight?
 - _Promise of generality_ The benefits of KNOs are touted, see e.g. the first sentence of the second paragraph of the introduction, yet the generality and performance seem to be in conflict. Consider the sentences
	"This explicit kernel representation allows desirable properties to be encoded directly into the kernel while avoiding restrictive assumptions, such as the radial, stationary, and periodic constraints inherent in FNOs. On regular grids, the KNO employs a fast dimension-wise factorization algorithm to mitigate the curse of dimensionality, ensuring computational efficiency in high-dimensional settings.''
	These two sentences undermine each other. The kernels _can_ be general, but in practice _need_ to be factorable otherwise they are unusable in higher dimensions. Of course, allowances for performance of practicality are usually needed, but a general architecture that must be heavily restricted is only superficially general.
 - _Reduced parameter count_ This is interesting, but not enough on its own. From Section 4.2,
	``For instance, with weights stored in fp32, the largest FNO model required approximately 54 MB of storage, the largest GNOT model required approximately 5.8 MB of storage, and the largest Transolver model required 15.2 MB of storage (all for the 3D NS Mach 1.0 problem). In contrast, the KNO required only 0.11 MB of storage for the same problem, highlighting its favorable memory scaling properties compared to the FNO, GNOT, and Transolver.''
	Indeed the KNO is smaller than competing architectures, but those architecture are quite small anyways. In short, a 54MB model is already very small. What unmet need is met by introducing models that are even smaller, especially if they are not faster?
 - _Numerical Performance_ The abstract explains
	``Numerical results demonstrate that on existing benchmarks the training and test accuracy of KNOs is comparable to or higher than popular operator learning techniques''
	A read of section 4.1 show that the test accuracy is sometimes worse than, sometimes higher than existing benchmarks. What's the story here?

Other questions:
 - In the first paragraph of section 2.3, it is not clear if the data is required to be on the same grid point. The text appears to imply one grid is used, but Appendix A.1.5 suggests that a separate training/evaluation grid is possible.
 - What does the relation in Eqn. 10 mean?
 - Just about Eqn. (9), is the phrase:
	"reduces the size of the kernel Gramians"
	What does the term "Gramians" mean in this context?

**Claims And Evidence:**

Yes

**Claims Explanation:**

I have checked the proofs of the theorems, they are correct. The numerical experiments appear correct as well.

**Requested Changes:**

The requested changes are, in short, to improve the significance of the work. This could be done by addressing the points above. Producing new theoretical results with more `surprise,' better demonstrating the value of a reduced parameter count, improved generality, etc.

---

> ### Author Response · Authors · 2026-03-04
> **Response to Reviewer nzrj**
>
> We thank the reviewer for their thoughtful response. We will attempt to address the reviewer’s comments carefully here:
>
> 1) Generalized architecture: we will address this below. See point 3.
>
> 2) Theoretical contributions: our primary contribution is not theoretical. The theorems are primarily there to show that the KNO can safely be used as a neural operator, since it is also a universal approximator. We hope to eventually follow up with more insightful theorems on, for instance, approximation rates and sampling (once we develop them), but that was not a focus of this work. We will edit the text to clarify this.
>
> 3) Promise of generality: We apologize for the lack of clarity in our writing; this has clearly caused unintended confusion. Our method is usable in higher dimensions without factorization, just like the FNO. We show exactly this in Appendix number A.7.6. However, on regular grids, we are able to speed things up considerably by factorization. Here, once again, our writing was ambiguous, but we would like to clarify: it is the integral operator that needs to be factorized, not the kernel. In fact, in general, our kernels are not factorized in any way. We will edit the text to make this clearer.
>
> 4) Reduced parameter count: This is a fair point, but we do still think that the KNO would be useful as an “on-chip” neural operator for devices with low memory storage (medical devices, for instance). In general, since the memory scaling of the KNO is more favorable, we anticipate being able to tackle large high-dimensional problems more easily. We will edit the text to add this point. In general, we find it scientifically interesting that a model with orders of magnitude fewer parameters is able to generally achieve errors close (on par, occasionally higher, usually lower) than other neural operators.
>
> 5) Numerical Performance: You’re right to point this out. Another reviewer also made a similar comment (see point 2 in our response to Reviewer VpVj). We will need to elaborate on this in the text. Essentially, while the errors induced by the KNO are a somewhat straightforward consequence of its individual pieces (quadrature, number of training points, channel dimension, and kernel expressivity), it is more challenging to analyze competing neural operators. However,
> we can safely state that the KNO is the second or first best performing model in all benchmark problems upon revisiting the NS-Pipe problem and the NS Mach 1.0 problem. In the former, we inadvertently used the wrong test set and the corrected result shows KNO as the best performing model; this KNO has a depth of 7, a ns-gsm kernel, and a channel dimension of 128.
>
>
>     Additionally, we performed an ablation study over model size on this dataset. In this study the fixed model had a ns-gsm kernel, a depth of 5 and a channel dimension of 64. Here one is able to achieve a more performant KNO by increasing depth or width, but depth adds expressivity with a smaller parameter cost.
>
>     ### NS-Pipe
>
>     | Model | KNO | Transolver (2nd best) |
>     |-|-|-|
>     | % NRMSE | 0.317 ± 0.015 | 0.33 ± 0.02 |
>     | Param count | 274,561 |2,810,817 |
>
>     | **Depth** |   4  |  5 |  6  |  7  |  8  |
>     |---------------------------|---------|---------|---------|---------|----------|
>     | % NRMSE   | 0.434 ± 0.0147  | 0.386 ± 0.025 | 0.401 ± 0.0210  | 0.352 ± 0.004 |  0.348 ± 0.021 |
>     | Param count | 61,057 | 74,177 | 87,297 | 100,417 | 113,537 |
>
>     | **Channels** |   16  |  32 |  64  |  128  |  192  | 256 |
>     |---------------------------|---------|---------|---------|---------|----------|----|
>     | % NRMSE   | 1.031 ± 0.018  | 0.579 ± 0.006 | 0.386 ± 0.025  | 0.360 ± 0.005 |  0.350 ± 0.007 | 0.343 ± 0.006 |
>     | Param count | 13,169 | 29,921 | 74,177 | 205,697 | 394,561 | 640,769 |
>
>
>     For the NS Mach 1.0 problem, KNO was able to perform much better after changing the normalization strategy to match Transolver's: from pointwise (per spatial location) normalization across training samples to global normalization using a single mean and standard deviation computed over the entire training set. So that the relative improvement of Transolver's result is 1.05%.
>
>     | **Model** | KNO| Transolver |
>     |-|-|-|
>     | % NRMSE | 48.636 ± 0.073 | 48.127 ± 0.483 |
>     | Param count | 31,105 | 3,791,377 |
>
>     Thus to paraphrase the amended results: For Burgers' problem, KNO is worse than solely FNO. For Beijing-air, Darcy (PWC) and NS-Pipe, KNO is the best performing model. For Darcy (triangle), the KM does the best, but KNO is equivalent to the FNO with regard to standard error. For NS Mach 1.0, KNO performs 1.05% worse than Transolver with 2 orders of magnitude less parameters, but better than all other methods. Finally, in React.-Diff. KNO outperforms all neural operators except KM.

---

> > ### Author Response · Authors · 2026-03-04
> > **Response to Reviewer nzrj continued**
> >
> > Response to other questions:
> >
> > 1) We apologize for the lack of clarity and will edit the text. Separate training and evaluation grids are indeed possible, much like in the FNO. We in fact illustrate evaluation on a finer grid in Appendix A.1.5.
> >
> > 2) The relation in Eqn 10 is stating that we approximate the $L^2_{\mu}$ norm of the operator (on the left) by the discretization on the right, with the input functions $f^{(m)}$ being sampled from $\mathcal{U}$ with the probability measure $\nu$. We will add a sentence below clarifying this.
> >
> > 3) The kernel Gramian is the matrix formed by evaluating the kernel at some set of points. That said, the kernel Gramian size only depends on the number of points and not their dimension, so we will remove that part of the sentence!

---

### Author Response · Authors · 2026-03-19
**Summary of revisions**

We thank the reviewers for their thoughtful and careful reviews. We have uploaded a revision with color-coded text.
1. We use blue for reviewer nzrj.
2. We use orange for reviewer gR1i.
3. We use red for reviewer VpVj.
In cases where reviewers gR1i and VpVj asked for similar or overlapping things, we used red for simplicity.

Here, we summarize the revisions made in response to reviewer comments:
1. The introduction (Section 1) and the connected work section (Section 1.1) have been rewritten to both incorporate a more complete literature review and to better motivate the need for low-memory neural operators (backed by citations to the literature).
2. As requested, Figure 1 now shows a top-level schematic of the KNO. Figure 2 shows a schematic of discretization of a single KNO layer, thereby summarizing almost the entire methods section.
3. Subsections 2.2-2.5 are now subsubsections with a new SubSection 2.2 titled "Discretization", which contains both Figure 2 and a top-level summary of the subsequent subsubsections. This should help reviewers and readers better navigate the density of that material.
4. A brief discussion on diagonal and low-rank integral operators and the choice of matrix-valued kernels for the KNO has been added to the new Section 2.21 (formerly Section 2.2). We also connect this to the appendix where we showed evidence for the benefits of non-diagonal matrix-valued kernels, but remark that diagonal is a reasonable choice for most problems (our results used diagonal kernels).
5. We added statements about when to expect the NS-GSM kernel to outperform the GSM kernel (and vice-versa) to both Section 2.2.1 and to the Results section.
6. In Section 2.2.1 on page 6, we clarified that the dimension-wise factorization is merely an efficiency improvement for regular grids, and that it does not require factorized kernels; rather, it is a factorization of the integral operator itself.
7. Right below that text near the bottom of page 6, we commented (in red) on how this obviates the need for FFT and has better scaling properties to high-dimensional problems.
8. We added clarifying sentences below equation 10, both about that equation and about the role of the interpolant. We remark that training and evaluation grids need not coincide.
9. We added clarifying language to multiple locations on the nature of the dimension-wise factorization, mentioning that it is merely an efficiency improvement that is made possible on regular grids, and that it is a factorization of the integral operator, not the kernel itself.
10. In Section 3, we added a statement describing why we present universal approximation theorems, and that we defer convergence rates and sampling results to future work. We also added the latter statement to the conclusion.
11. We updated our results as described in the conversation with the reviewers. The KNO is now the best or second-best neural operator across all results. When second best, it is extremely close in accuracy to the best performing operator (mostly indistinguishable). We also made sure to compare our results to the ensemble work of Sharma and Shankar, as requested by the reviewer.
12. In Section 4.2 on parameter counts, we updated the text to note that we typically trained and inferred faster than the Transolver and GNOT, likely due to our significantly smaller parameter counts. We added clarifying language on why we believe this is important, mirroring similar statements in the introduction.
13. We updated the conclusion to discuss Warp and Triton, and described paths to speeding up the KNO through efficient implementation.
14. We updated the conclusion to discuss extensions to sparse grids, Monte Carlo rules, etc in high dimensional settings.
15. We also updated the appendix to add an ablation on the NS-pipe example (Table 15) and to update the standard errors table (Table 8) to include the NS Mach 1 and NS Pipe example updates that were folded into the main results.

We generally note that the KNO's architecture allows for very seamless integration into existing scientific computing pipelines, since it is built out of similar ingredients (quadrature, closed-form kernels, interpolation). A sentence to this effect is now in the introduction.

---

### Decision · Action_Editor_vyBb · 2026-04-10

**Recommendation:** Accept as is

**Additional Comments:**

Reviewer gR1i mentions:  "that some typos still remain (for instance, a few added citations seem to be repetitive). The authors are strongly encouraged to resolve these typos later on."

Please, address these coments in the final version.

**Audience:**

Yes

**Audience Explanation:**

The paper is likely to interest at least part of the TMLR audience, especially researchers in operator learning, scientific machine learning, and PDE-based ML. Two reviewers explicitly affirmed audience interest, citing the relevance of explicit kernels and the connection to current AI for science research. Although one reviewer initially expressed doubts about significance and novelty, that reviewer later recommended acceptance after revision. In addition, the paper’s emphasis on memory efficiency and geometric flexibility makes it practically relevant for researchers concerned with scalability and deployment in scientific computing settings.

**Claims And Evidence:**

Yes

**Claims Explanation:**

All three reviewers confirm the correctness of the theoretical results (universal approximation theorems) and the numerical experiments. Initial concerns about numerical performance were resolved after the authors corrected some of the experiments and provided additional ablation studies. The proofs have been verified as correct, and experimental details are thoroughly documented.

---

> ### Author Response · Authors · 2026-04-11
>
> Dear Action Editor,
>
> Thank you, this is great news! We will address these final comments and upload asap.

---

> > ### Author Response · Authors · 2026-04-29
> > **Official Comment by Authors**
> >
> > Dear Action Editor,
> >
> > We have uploaded the camera ready version of the paper after addressing the typos and repetitive citations.